# Real-Time Monitoring of Cable Sag and Overhead Power Line Parameters Based on a Distributed Sensor Network and Implementation in a Web Server and IoT

**DOI:** 10.3390/s24134283

**Published:** 2024-07-01

**Authors:** Claudiu-Ionel Nicola, Marcel Nicola, Dumitru Sacerdoțianu, Ion Pătru

**Affiliations:** 1Research and Development Department, National Institute for Research, Development and Testing in Electrical Engineering—ICMET Craiova, 200746 Craiova, Romania; 2Department of Automatic Control and Electronics, University of Craiova, 200585 Craiova, Romania

**Keywords:** cable sag, ZigBee, ModBUS, web server, IoT

## Abstract

Based on the need for real-time sag monitoring of Overhead Power Lines (OPL) for electricity transmission, this article presents the implementation of a hardware and software system for online monitoring of OPL cables. The mathematical model based on differential equations and the methods of algorithmic calculation of OPL cable sag are presented. Considering that, based on the mathematical model presented, the calculation of cable sag can be done in different ways depending on the sensors used, and the presented application uses a variety of sensors. Therefore, a direct calculation is made using one of the different methods. Subsequently, the verification relations are highlighted directly, and in return, the calculation by the alternative method, which uses another group of sensors, generates both a verification of the calculation and the functionality of the sensors, thus obtaining a defect observer of the sensors. The hardware architecture of the OPL cable online monitoring application is presented, together with the main characteristics of the sensors and communication equipment used. The configurations required to transmit data using the ModBUS and ZigBee protocols are also presented. The main software modules of the OPL cable condition monitoring application are described, which ensure the monitoring of the main parameters of the power line and the visualisation of the results both on the electricity provider’s intranet using a web server and MySQL database, and on the Internet using an Internet of Things (IoT) server. This categorisation of the data visualisation mode is done in such a way as to ensure a high level of cyber security. Also, the global accuracy of the entire OPL cable sag calculus system is estimated at 0.1%. Starting from the mathematical model of the OPL cable sag calculation, it goes through the stages of creating such a monitoring system, from the numerical simulations carried out using Matlab to the real-time implementation of this monitoring application using Laboratory Virtual Instrument Engineering Workbench (LabVIEW).

## 1. Introduction

Optimising the operation of high-voltage overhead lines and ensuring their operational safety is essential, whether the power lines are old or new, and regardless of weather conditions. The most modern solution for controlling and optimising the operation of overhead power lines requires the use of intelligent networks based on systems for on-line monitoring of operating parameters and conditions. In the context of smart grids, energy efficiency refers to reducing the congestion and operational losses of high-voltage overhead lines, increasing their operational safety and reducing the need for investment in new overhead lines. To maximise energy efficiency, it is necessary to ensure the controllable and efficient operation of OPL of different voltages [1,2].

An intelligent electricity transmission network should have, in addition to energy-efficient power lines, on-line control and monitoring systems necessary to achieve efficient programming of the operation and coordination of electricity transmission needs in relation to the technical condition of power lines and primary electrical equipment in high-voltage power stations, as well as environmental conditions. In the process of transporting electricity through electrical networks, the quality of the cable essentially sets the limit of the load capacity. To prevent overloading, the designed nominal load capacity of the transmission line is set as a fixed value quantity that considers the most unfavourable meteorological conditions. However, normally, such unfavourable weather conditions are rare, so the transmission potential of cable power is not used to its maximum in most cases [3,4].

The need to monitor and assess the condition of power transmission lines in real time as accurately as possible is obvious. The problem, then, is to develop specific and efficient calculation algorithms based on the descriptive equations for calculating cable sag, and subsequently for real-time implementation, to use a set of suitable sensors together with a communication system that allows the data to be transmitted to a central server that performs the usual operations of displaying the state of the monitored line, but also integrates with the local Supervisory Control and Data Acquisition (SCADA) system [5,6,7,8].

As a result, several systems have been developed to monitor the condition of power transmission lines, where the calculation of cable sag is of particular importance. Among the specific elements presented in the scientific literature, we can mention: monitoring based on a network of wireless sensors [9,10], the creation of a data communication system based on Long Range (LoRa) [11], the monitoring of power lines using chirped fibre Bragg grating [12,13], and the realisation of a power line monitoring system based on Differential Global Positioning System (DGPS) [14].

This article is a continuation of [15] and presents the implementation of a hardware and software system for online monitoring of OPL cables and the determination of cable sag. Starting from the mathematical model of the OPL cable sag calculation, it goes through the stages of creating such a monitoring system, starting with the numerical simulations carried out using the Matlab R2018b programming environment [16,17] and continuing with the real-time implementation in the LabVIEW 2015 programming environment [18,19] of this monitoring application. In the paper presented, the line slope method has been used in principle, where based on the detection of the OPL cable slope angles, the sag value can be achieved to obtain an alternative verification relation, to which the measurement of the mechanical cable stress is also used, thus achieving increased accuracy and redundancy without significant burden on the hardware system. Table 1 summarises the methods, advantages and disadvantages of previous work.

The main modules of the application software are: the software module for reading quantities from the sensor network through a ModBUS Input/Output (I/O) server, the software module for real-time calculation of the OPL cable sag, the software module for read/write operations based on MySQL type data, the software module for transferring data to the user’s intranet through a web server, and the software module for viewing data by authorised users based on user and password through an IoT server.

The logical chain of the software elements is based on the Matlab environment for the numerical simulations, while for the real-time implementation, the LabVIEW development environment is used, which is suitable for the global integration of the system including Matlab scripts for calculations, but also the integration of a MySQL database, a web server and an IoT server. Thus, the main contributions of this article are as follows:Establishing alternative calculation methods and verification calculation relations for the OPL cable sag calculation depending on the groups of sensors used, thus obtaining a software method for detecting the failure of a sensor, a method whose usefulness can be expressed in terms of increasing system reliability;Establishing the hardware and software architecture of the OPL online cable monitoring application;Setting up the necessary configurations for data transmission using ModBUS [20] and ZigBee [21] protocols in order to minimise energy consumption in the process of local centralisation of data from sensors and transmission via Global System for Mobile Communications (GSM) [22] to the user server;Implementation of the software module for reading quantities from the sensor network through a ModBUS I/O server;Implementation of the software module for the real-time calculation of the OPL cable sag and monitoring of the parameters of interest of the power line, including the correct functioning of the sensors and maintenance within the acceptance limits of the OPL cable sag values;Implementation of a software module to send automatic e-mails with warnings and alarms regarding OPL cable sag limits and sensor faults;Implementation of the software module for the transfer of data to the user’s Internet via a web server [23,24];Implementation of the software module for viewing data by authorised users, based on user and password, via an IoT server [25,26].

The rest of the article is structured as follows: Section 2 presents the cables and the calculation. The general architecture of the overhead power line cable monitoring system is presented in Section 3, and software applications for the overhead power line cable monitoring system and cable sag calculation are presented in Section 4. Some conclusions and suggestions for further developments are presented in the final section.

## 2. Cable Sag Calculation

This section presents the method of calculating the sag of the power cable of overhead power lines, the simplified geometry of which is shown in Figure 1. The location of the *xOy* plane coordinate system at the base of the left pole is observed, and the main notations used are presented below. The angles formed by the tangent of the cable at the ends with the axis of the *Ox* are denoted by *θ*_0_ and *θ*_1_, whose measure is given in degrees, and the difference between the poles, the cable rise *h*, is measured in metres. The weight of the cable, measured in N, is written as *G*, the weight/length of the cable, measured in N/m, is written as *w*, and the length of the cable, *l*, is measured in metres. Also, the projection of the mechanical cable stress on the *Ox* axis (horizontal component) is denoted by *H*, and (*x**, *y**) are the coordinates of the lowest point of the cable measured on the *Oy* axis, as shown in Figure 1 [15,27].

To calculate the sag of the cable in the structure of the overhead power lines, the catenary equation is used, which can be derived in a simplified way from Figure 2. Thus, in Figure 2, using the usual notation, an infinitesimal part of the cable is shown, which is considered to be inextensible in a first approximation, after which the following sections describe certain corrections that can be made to increase the accuracy of the calculations because, in reality, the cable will undergo contraction or expansion as a result of variations in the ambient temperature or the electrical load on the line. Figure 2 also shows the vector decompositions of the main forces that occur in the cable where the gravitational load is applied.

The balance equations can be written on the horizontal and vertical axes in Equations (2) and (3) using notations (1), where s is the coordinate of the length of the cable.
(1)$$\left\{ \begin{array}{l} \sin(\theta )=\frac{dy}{ds}\\ \cos(\theta )=\frac{dx}{ds}\\ \tan(\theta )=\frac{dy}{dx} \end{array}\right.$$

For the equilibrium on the vertical axis *Oy*, we get the following:(2)$$\frac{d}{ds}\left(T\frac{dy}{ds}\right)=mg$$

For the equilibrium on the horizontal axis *Ox*, we get the following:(3)$$\frac{d}{ds}\left(T\frac{dx}{ds}\right)=0$$

In relations (2) and (3), the mechanical stress in the cable is denoted as $T=f_{T}(s)$. By integrating Equation (3), we obtain Equation (4).
(4)$$T\frac{dx}{ds}=H$$

In Equation (4), *H* is the integration constant and, as there is no horizontal force acting on the cable, it is equal to the horizontal component of the mechanical cable stress.

From Equation (4), we know that $T=\frac{ds}{dx}H$, and noting w=mg (the weight per unit length) from Equation (2) is obtained after multiplying both members with the following equation:(5)$$H\frac{d^{2}y}{dx^{2}}=w\frac{ds}{dx}$$

Given the fact that *ds* is infinitesimal, the following geometric relation can be written:(6)$${\left(ds\right)}^{2}={\left(dx\right)}^{2}+{\left(dy\right)}^{2}$$

Based on this, Equation (5) can be described in the form:(7)$$H\frac{d^{2}y}{dx^{2}}=q_{c}\sqrt{1+{\left(\frac{dy}{dx}\right)}^{2}}$$

To simplify the writing of the equations, the following notation is introduced $a=\frac{dy}{dx}$. Thus, the following equation is obtained
(8)$$\frac{da}{dx}=\frac{w}{H}\sqrt{1+a^{2}}$$

Integrating relation (8), the following relation is obtained:(9)$$\int \frac{da}{\sqrt{1+a^{2}}}=\int \frac{w}{H}dx\Leftrightarrow \ln\left(a+\sqrt{1+a^{2}}\right)=\frac{w}{H}x+K_{1}$$
where *K*_1_ represents an integration constant.

Next, the substitution $b=\frac{w}{H}x+K_{1}$ is made and by exponentiating the relation (9), the following relation is obtained:(10)$$a+\sqrt{1+a^{2}}=e^{b}\Leftrightarrow \sqrt{1+a^{2}}=e^{b}-a$$

Squaring Equation (10), and subsequently solving for *a*, results in the following form:(11)$$a=\frac{e^{b}-e^{-b}}{2}=\sinh(b)\Leftrightarrow \frac{dy}{dx}=\sin\left(\frac{w}{H}x+K_{1}\right)$$

The integration of Equation (11) gives the following form:(12)$$y=f_{y}(x)=\int \sinh\left(\frac{w}{H}x+K_{1}\right)dx=\frac{H}{w}\cosh\;\left(\frac{w}{H}x+K_{1}\right)+K_{2}$$
where *K*_2_ represents another integration constant.

The differential form of Equation (12) can be written as follows:(13)$$y^{\prime }=\frac{dy}{dx}=\sinh\left(\frac{w}{H}\cdot x+K_{1}\right)$$

The expressions of the two integration constants *K*_1_ and *K*_2_ are given in the following form [15,27]:(14)$$\left\{ \begin{array}{l} K_{1}=a\sinh\left(\frac{w\cdot h}{2\cdot H\cdot \sinh\frac{w\cdot L}{2\cdot h}}\right)\\ K_{2}=-\frac{H}{w}\cdot \cosh(K_{1}) \end{array}\right.$$

The following relationship can be used to calculate the length of the cable:(15)$$l=\int_{0}^{L}\sqrt{1+{\left(\frac{dy}{dx}\right)}^{2}}dx,$$

After a series of calculations, Equation (15) explicitly gives the length of the cable:(16)$$l=\frac{H}{w}\left(\sinh\left(\frac{w\cdot L}{H}+K_{1}\right)-\sinh(K_{1})\right)$$

Because $w=\frac{G}{l}$, the coordinates *x** and *y** are expressed in the form [15,27]:(17)$$x^{*}=\frac{-H\cdot K_{1}}{w}$$
(18)$$y^{*}=\frac{H}{w}\cosh\left(\frac{w\cdot x^{*}}{H}+K_{1}\right)+K_{2}$$

The following expression can also be written from Figure 1:(19)$$H=\frac{G}{\tan(\theta_{0})+\tan(\theta_{1})}$$

Equation (19) can be used as an alternative calculation equation, given that *x** and *y**, which characterise the length of the arrow, can be calculated according to the horizontal mechanical stress, which can be provided directly by a mechanical stress sensor, or with the help of two inclination sensors providing the angles *θ*_0_ and *θ*_1_ from the Equation (19). Depending on the economic constraints, the system can be equipped with either one or both options for determining the horizontal mechanical stress *H*. It can therefore be concluded that by verifying the complementary relation to determine the horizontal mechanical stress *H*, the functional state or fault of the sensors involved can be directly checked when calculating the cable sag value. The result is a software-based method of detecting sensor faults, which can increase system reliability.

For the case of *x_A_* = 0, *y_A_* = 0, *x_B_* = *h*, *y_B_* = *L*, Equations (13) and (14) can be found as a graphical representation in Figure 1.

According to [15,27], after determining the points (*x**, *y**), in addition to checking the correctness of the calculations for the cable sag value, the following relation must also be checked:(20)$$0=-y_{A}+\frac{H}{w}\cosh\;\left(\frac{w}{H}(x_{A}-x^{*})\right)-\frac{H}{w}+y^{*}$$
(21)$$0=-y_{B}+\frac{H}{w}\cosh\;\left(\frac{w}{H}(x_{B}-x^{*})\right)-\frac{H}{w}+y^{*}$$

With these, the method of calculating cable sag can be synthesized as given in Algorithm 1. The implementation of this algorithm was done in the Matlab environment. Starting from the geometrical data of the line section and measuring the horizontal mechanical stress in the cable, the algorithm involves solving an implicit equation to find the cable length *l* and then the cable sag.

**Algorithm 1** Matlab implementation for cable sag calculation
1. h = **input**(‘Give the cable rise h [m]:’);2. L = **input**(‘Give the distance between the pillars L [m]:’);3. G = **input**(‘Give the cable weight G [N]:’);4. H = **input**(‘Give the mechanical cable stress H [N]:’);5. T = (G × 10)/H;6. y = **@**(l)(T − sinh(asinh((h × T)/(2 × l × sinh((L × T)/(2 × l)))) + ((L × T)/(2 × l))) + …7.   sinh(asinh((h × T)/(2 × l × sinh((L × T)/(2 × l)))) − ((L × T)/(2 × l))));8. a = **fsolve**(y,L);          % Plot the function9. w = (10 × G)/a;10. K1 = **asinh**((w × h)/(2 × H × sinh((w × L)/(2 × H)))) − (w × L)/(2 × H);11. K2 = −(H/w) × **cosh**(K1);12. z = **@**(x)((H/w) × cosh((w/H) × x + K1) + K2);13. **fplot**(z,[0 L]);          % Plot the function14.                 % Find and plot the minimum15. **minimum** = **fminbnd**(z,0,L);    % This function handle directly to the minimization routine16. **plot**(minimum,z(minimum),‘d’); % Evaluate the function without using feval17. z(minimum)18. x_s = (−H × K1)/w19. y_s = (H/w) × cosh((w × x_s)/H + K1) + K2


Therefore, for the numerical simulations presented below, the sag calculation for the OPL cables is used as a first example for a set of data presented in Table 2.

Figure 3 shows the result of the numerical simulation for the graphical representation of the determination of the OPL cable sag value for the input data given in Table 2, using the algorithm performed in Matlab.

Table 3 shows the numerical simulations for another set of data in the case where the two pillars are positioned differently on the *Ox* axis.

For the input data given in Table 3, Figure 4 shows the result of the numerical simulation for the graphical representation of the determination of the OPL-cable sag value using the same algorithm made in Matlab.

## 3. General Architecture for Overhead Power Line Cables Monitoring System

The implementation of the proposed hardware system and software applications for data acquisition and transmission for the online monitoring system of OPL cables is based on the ModBUS and ZigBee protocols. The ModBUS protocol is a communication protocol based on a master–slave or client–server architecture, and the main purpose of the ModBUS protocol is to facilitate stable and fast communication between the slave hardware subsystem in the ZigBee wireless network and the sensors in the subsystem component OPL cable monitoring hardware. ZigBee, as described in the IEEE 802.15.4 standard [28], is a low-bitrate data transfer protocol for Wireless Personal Area Networks (WPAN). It is designed for easy connection between devices, keeping power consumption to an optimum minimum. The ZigBee network is self-organising, requiring minimal user or administrator intervention during initial configuration. Further intervention is only required in major problem situations where a very large number of nodes fail, or where running configurations are deleted and reset [20,21].

The hardware/software configuration for data transmission to monitor, analyse and diagnose OPL cables is presented in Figure 5. Also, the proposed and realized variants of the communication architectures for “Analog value Modbus RTU stage” and for “Modbus RTU-ZigBee” are not fully redundant, and the purpose of the introduced redundancy is also based on a study of the requirements of the potential beneficiaries (electricity transmission companies).

Each sensor can be parameterised using a hardware/software module that interfaces between the sensor and the general software application using the ModBUS communication protocol. Each sensor is therefore assigned an ID and the read/write functions specific to the ModBUS communication protocol make it possible to read the data provided by the sensors in real time, as well as their parameterisation.

The software applications developed for the online monitoring subsystem of the OPL cables are implemented using the ModBUS protocol and involve writing software functions for each sensor to set minimum and maximum values, various thresholds and state values. According to Figure 5, the data acquired by the network of sensors presented in the previous section are collected in a unified manner on the ModBUS protocol in a local gateway Teltonika TRB 145 type, which will transmit them to the application server of the OPL cable monitoring system on the 3G/4G protocol.

In Figure 6, groups of some images represent aspects of the implementation phase of the hardware modules to realize the OPL cables monitoring system.

Thus, Figure 6a shows the measurement sensors that provide the instantaneous status of the cable, namely: temperature, tilt, vibration, communication module for integration into the ZigBee network, and a specialized electronic circuit that performs filtering and voltage stabilization of the components. Figure 6b presents the power supply system of the cabinet located on the OPL cables consisting of two split core current transformers SCT045B-type, which transform the energy radiated by the electric cable on which it is mounted (up to 1000 A and 400 kV), at the output of the transformer obtaining a power of 10 VA. Also, Figure 6b shows the sensor for measuring the value of the current through the cable represented by a Hall split-core current transducer. The power system of the cabinet located on the pillar consists of 200 W solar panels and 153 Wh buffer battery. Figure 6c shows the implementation of the power line pillar hardware module cabinet with the following: tilt sensor, accelerometer sensor, signal conditioner for mechanical stress sensor, controller for photovoltaic panels, ModBUS RTU to ZigBee coordinator, and GSM gateway. The climatic conditions sensors (anemometer, temperature, humidity, wind, hail, and ice) implementation is shown in Figure 6d.

### 3.1. Overhead Power Line Cables Hardware Monitoring Cabinet

Described below, the cabinet for the OPL cables hardware monitoring module based on the architecture shown in Figure 5 and with the implementation shown in Figure 6a,b has the following components in its structure.

The power supply system is represented by two split core current transformers SCT045B-type PowerUC Electronics, Piastów, Poland, which transform the energy radiated by the electric cable on which it is mounted (up to 1000 A and 400 kV), at the output of the transformer obtaining a power of 10 VA.

To measure the current through OPL cables, a Hall split-core current transducer type THST45A is used, which provides a measuring range of up to 1200 A. The analogue output of this type of transducer is 0 ÷ 5 V.

The inclinometers used to measure the pitch and roll angles of the OPL cables are the DAS-90-A from Level Developments [29]. They use a micro-electro-mechanical systems (MEMS) sensor in a rugged, sealed aluminium structure with a shielded polyurethane (PUR) cable and a 4-pin M12 connector. The sensor is powered in the range 7 ÷ 32 VDC and the measured angle is in the range −90° to 90°. The sensor output is analogue 0.4 ÷ 5 V, and the measurement accuracy is ±0.34°. The sensitivity for the first measuring interval of 1° is 35 mV/° and for the interval 0 ÷ 1 g it is 2 V/g. Concerning the change in sensitivity of the sensor with temperature, an error correction is applied, given by the relation:(22)$$E_{sd}=SD\times \Delta T\times \theta$$
where *E_sd_* is the output variation in degrees corresponding to the sensitivity with temperature, *SD* is the significance corresponding to the ±0.003%/°C drift, ∆*T* is the temperature change interval in °C, and *θ* is the current angle in degrees (°) and is provided by the inclinometer (tilt) sensor. Similar to the correction Equation (22), each sensor allows for the implementation of similar equations in the OPL cable calculation software application.

The PT100 sensor is a temperature sensor for the range −50 ÷ 400 °C with an accuracy of 0.2% and a drift of less than 0.04%. The material used to manufacture this type of sensor is tetrafluoride silver-plated wire and stainless steel. The conversion to the analogue quantity 0 ÷ 5 V is provided by a module type HIC-PT100.

The Recovib IAC-CM-U-03 sensor from Micromega Dynamics [30] is used to measure OPL cable vibration. It has a measurement sensitivity of 40 mV/g over a range of −50 to 50 g and a drift offset of ±50 mg/°C. This sensor provides an analogue output of 0 to 5 V.

The sensors presented above are connected to the ICP DAS tM-AD8 hardware module, which converts analogue inputs into digital outputs on the ModBUS RTU protocol on the RS-485 interface. The command line tool software interface can be used to test and debug the calibration and configuration modules using ModBUS Remote Terminal Unit (RTU) and DCON protocol commands. Figure 7 shows the basic configuration of the analogue input to ModBUS RTU hardware module.

Data transmission between the two cabinets located on the respective line on the pole is achieved using the ZT-2550 and ZT-2551 series modules, which are small, wireless ZigBee converters based on the IEEE802.15.4 standard, which allows the RS-232 and RS-485 interfaces to be converted into a ZigBee Personal Area Networks network (PAN) [21,22]. The typical transmission distance of ZT series ZigBee devices is 700 metres, with a transmission frequency range of 2.405 GHz to 2.48 GHz, divided into 5 MHz sectors, providing 16 channels and 16,384 PAN IDs. The ZT-2000 Series is not only a long-range wireless converter, but it can also act as a ZigBee router to extend the range and improve the quality of the wireless signal. In a ZigBee network, there is only one ZigBee host, known as the ‘ZigBee Coordinator’, and ZT-2550 Series devices are used to initiate and control routing. Furthermore, a single ZigBee network can manage up to 255 ZigBee routers and is responsible for receiving or bypassing data. Features: Industrial, Scientific, and Medical (ISM) 2.4 GHz operating frequency, compatible with IEEE802.15.4/ZigBee, adjustable RF transmission output power, support AES-128 encryption for wireless communication, RS-232/RS-485 interface, source identification for data transmission of device without address, supports broadcast transmission for redundant transmission path, supports unicast transmission to reduce network load and supports topology utility for network monitoring and improvement. The ZT-2000 Series Topology Utility configuration software is used to configure ZigBee networks. Thus, the ZT-2551 converter-type module is used in the OPL cables cabinet.

### 3.2. Overhead Power Line Pillar Hardware Monitoring Cabinet

The interface and software configuration of the other sensors, i.e., the OPL cable mechanical force sensor, the pillar inclination measurement sensor, the vibration sensor and the sensors included in the weather station installed on the pillar, as well as their integration into the ModBUS network for data transmission to the server, is carried out in the same way as described above. The cabinet of the OPL pillar hardware monitoring module based on the architecture shown in Figure 5 and with the actual implementation shown in Figure 6c,d has the components described below in its structure.

The sensor that measures the inclination of the pillar is of the SOLAR-2 type from Level Developments. The sensor is powered in the range 9 ÷ 30 VDC, and the measured angle is in the range −45° to 45°. The sensor output is ModBUS type and the measurement accuracy is ±0.040° at a temperature of 10 °C. The sensitivity for the first measurement interval is ±0.0006%/°C and the correction for the change in sensitivity of the sensor with temperature is given by Equation (22).

The accelerometer-type sensor used to measure the vibration of the OPL pillars is the HWT605-485 from WIT Motion and is a device with an Inertial Measurement Unit (IMU) sensor [31]. This type of sensor measures accelerations in the range of −6 g to 6 g and has a measurement accuracy of 0.01 g. The sensor integrates into the local ModBUS communication network with the default ID 0x50.

A versatile load cell, the DCL10 Central Load Cell, is used to measure mechanical stress in the cable. This nickel-plated sensor is manufactured from alloy steel and is IP67-rated for outdoor use. The non-linearity is 0.03% of the full-scale value. A signal conditioner provides the measured value both in a unified 4 ÷ 20 mA; 0 ÷ 10 V signal and as a ModBUS protocol on the RS-485 interface. By default, on the local ModBUS network, the sensor ID is 0x04, and the address of the provided size is 57 H and 58 H [32].

In the structure of the cabinet located on the line pillar, the ZT-2550 coordinator module is used, which takes the data from the ZT 2551 converter module in the cabinet located on the line via the ZigBee protocol and transmits them via the ModBUS protocol to the GSM gateway hardware module described below.

Thus, the gateway board Teltonika TRB145 the industrial GSM/General Packet Radio Service (GPRS) is used, which is equipped with an RS4-85 interface, digital inputs/outputs and a micro USB port. It has a wide range of software features including SMS control, firewall, OpenVPN, IPsec, RMS and FOTA support. Hardware features: Mobile LTE (Cat1)/3G/2G, SIM slot Mini SIM (2FF), 1xSMA antenna connector for GSM, RS485 Full Duplex (4 wire) and Half Duplex (2 wire), transmission speed 300 ÷ 115,200, input/output 2xI/O digital/analog (configurable), CPU ARM Cortex-A7 CPU 1.2 GHz, RAM-type memory 128 MB, flash-type memory 512 MB, power supply 9 ÷ 30 VDC, power consumption 5 W, and configuration interface: virtual NIC via Micro-USB connector. Software Features: Linux OS with Yocto SDK, Network NAT, Firewall, Dynamic Domain Name System (DDNS), RS-485 ModBUS, Serial over Internet Protocol (IP), and Networked Transport of Radio Technical Commission for Maritime Services (RTCM) over Internet Protocol (NTRIP) related services [33].

## 4. Software Applications for Overhead Power Line Cables Monitoring System and Cable Sag Calculation

This section presents the architecture of the software application that allows the calculation of the OPL cable sag based on the measurements provided by the sensor network, together with the storage of the results in a database to create a history, as well as the presentation of the data on the intranet through a web server or on the Internet through an IoT. The application software also allows integration with the user’s SCADA system, which is usually that of an electricity wholesaler. SCADA is one of the modern tools used to control and monitor technological processes. The main components of SCADA systems are one or more servers and clients. The servers used in this system are of the Open Platform Communications (OPC) type, which aims to define a common interface that requires only one design stage and then multiple reuses for any other SCADA project or other software packages [23,24,25].

Next, this section presents the software application, based on an NI-OPC server provided by National Instruments, which can be integrated into a SCADA-type system, and performs the following functions: the process of monitoring the state of the OPL cables and the management of client-server communication of quantities through online viewing and creation of records using TDMS (Technical Data Management Streaming) type files. A MySQL database server is used with data write/read applications from the OPL cables monitoring process. The data from the OPL cable condition monitoring process is displayed over the intranet/internet network via a web server embedded in the application, along with the presentation of a software application that transmits the data to an IoT-type cloud platform.

The software application for the OPL cable condition monitoring process is developed in the LabVIEW graphical programming environment, a programming environment that incorporates all the concepts listed above and the most common industrial process control techniques. The proposed software architecture of the OPL cable monitoring system is shown in Figure 8.

The software application modules of the OPL cable condition monitoring process implemented on the server computer are shown in Figure 8, and are as follows:The LabVIEW software module application client for reading values from the sensor network via a ModBUS I/O server;Matlab and LabVIEW software module application for real-time calculation of the OPL cable sag;LabVIEW software module client application for writing to and reading from the MySQL database;LabVIEW software module client application to send automatic e-mails with warnings and alarms about OPL cable sag limits and sensor faults;LabVIEW software module client application for data transmission on the electricity wholesaler’s intranet via a web server;LabVIEW software module client application for viewing data by authorised users, based on username and password, through an IoT server.

National Instruments provides three main mechanisms for interfacing with ModBUS devices: an OPC server, a ModBUS I/O server, and a ModBUS Application Programming Interface (API) integrated into the LabVIEW programming environment through LaVIEW Real-Time or the LabVIEW Datalogging and Supervisory Control (DSC) modules. Modbus I/O servers provide a high-level engine for ModBUS communication, and to use this type of server, a new I/O server is added from the LabVIEW library to the target built in the application project [18,19,24].

Figure 9 shows the data transfer flow used in the implementation carried out in the LabVIEW software project based on the ModBUS I/O server to read real-time data from the sensor network (cable and pillar hardware modules).

Also, LabVIEW and Matlab software block diagram of the OPL cable sag value real-time calculus application software module is shown in Figure 10.

A MySQL Server type database server is used to store the data to create the data log. It is responsible for storing data, finding it efficiently based on queries defined by the client, but also for handling the different levels of security required in the normal work with this data. MySQL is an open-source database server. It is a relational database management system based on the client-server model using the SQL language. MySQL’s features include: multi-threaded and multi-processor operation, functionality on a wide range of operating systems, support for large databases and a high level of security based on a flexible and secure system of passwords and privileges [18,23,24,25]. The use of Open Database Connectivity (ODBC) makes it easy for the OPL cable condition monitoring process application program to write to and query the MySQL server database. Figure 11 shows the configuration of the ODBC connection between the software application developed in LabVIEW and the MySQL server database.

The workflow for writing data to the MySQL database is the following: open connection, execute your query, fetch data, and close connection. Thus, Figure 12 presents the mentioned stages, which are: the stage for opening MySQL and LabVIEW ODBC connection; while looping the stage of inserting or reading data from LabVIEW monitoring local client to MySQL database; and the stage for closing MySQL and LabVIEW ODBC connection.

The system can also automatically send warning e-mails to pre-defined addresses if the cable sag value is outside the pre-defined safety limits, or if a fault is detected in one of the sensors because of the checks carried out in parallel with the cable sag value, as described in Section 2. The block diagram of the automated sending warning email from OPL cable monitoring is presented in Figure 13.

Normally, the results of the OPL cable condition monitoring software application are visible on the computer of the intended user, or in this case, an electricity transmission operator, but by using a web server, the data can be viewed within the operator, thus ensuring a higher level of security for the application. Figure 14 shows the block diagram of the web server application with data from OPL cable monitoring. Thus, a web client can exchange data with a stand-alone application developed in LabVIEW over a network (Intranet/Internet) via LabVIEW Web Services. Such a Web service consists of Virtual Instruments (VIs) and other files running on a server that responds to Hypertext Transfer Protocol (HTTP) requests from clients. A Web client, such as a browser, exchanges data with a Web service by sending an HTTP request to a specific Uniform Resource Locator (URL) address, whereas the LabVIEW embedded web server maps a URL to each HTTP VI method so that the specific URL used by the client determines which HTTP VI method receives and resolves the HTTP request. The request contains values for assigning specific parameters in the HTTP GET method, and after each request, the HTTP GET method processes these values and returns a response [24,25].

For greater flexibility in sharing data from the OPL cable monitoring process, an IoT context can also be used as in Figure 15. This ensures that data can be viewed, analysed and processed in the cloud by authorised users who have been securely issued with a username and password by the central user within the electricity provider.

## 5. Results and Discussion

As for the testing of the proposed monitoring system shown in Figure 16, it was done under quasi-real conditions in our laboratory (High Voltage Laboratory), where the maximum distance between two pillars is less than 40 m and the height is up to 30 m.

### 5.1. Experiment Description

For example, for the case of tilt sensors data acquisition, Figure 17 shows how the LabVIEW software block diagram application is implemented via the ModBUS I/O server for cable tilt angle monitoring. Similarly, data is acquired from all sensors within the two cabinets, as shown in Figure 9. Additionally, by using the algorithms developed in Matlab and LabVIEW, presented in Section 2, the real-time calculation of the cable sag value is performed.

Figure 18 shows the LabVIEW software block diagram of the application client software module for writing to the MySQL server-type database and the graphical interface of this process. Thus, this figure presents the usual stages of this process, namely, opening the MySQL server and LabVIEW ODBC connection, inserting data into the MySQL database, and closing the connection.

Table 4 presents the experimental data acquired and stored on the MySQL Database Workbench interface as follows: cable left tilt [°], cable right tilt [°], cable run *L* [m], cable weight *G* [N], cable rise *h* [m], cable mechanical stress *H* [kN], cable weight/length *w* [N/m], and (*x**, *y**) sag values point. In this experiment, the cable run is 30 m, which is the distance between two pillars, and the cable weight is 50 Kg.

An example of selecting experimental data from the MySQL server database using the read client software module application program written in LabVIEW is shown in Figure 19. Thus, Figure 19a presents the block diagram of the implementation for the read client software module application. The software interface displaying the result of the MySQL database selection is shown in Figure 19b. Additionally, on the interface in Figure 19b, two indicators are positioned, which inform the user that the cable sag value is within the normal range for the section of line on which it has been placed, and also after reading all the sensors, it is displayed that the status of the sensors is appropriate.

In Figure 20, by utilizing the automatic ActiveX functions such as a server and support for ActiveX Containers and ActiveX Events to employ the Simple Mail Transfer Protocol (SMTP) service for the Gmail-type e-mail server [24], the software block diagram demonstrates the client application for generating automated e-mails.

The web service is configured to return data in JavaScript Object Notation (JSON) format, which makes it very easy to load it as an object in JavaScript. This software procedure is performed by the JSON.parse function, which is available in Web browsers. JSON is very similar to cluster data in the LabVIEW programming environment. In a JSON object, each context has a key. In a LabVIEW cluster, each element can have a name, making it easy to modify clusters using the unbundle/bundle by name functions. This application will use a function that converts LabVIEW clusters to JSON objects. The function converts the element name of each cluster into a JSON key. Clients use URLs and HTTP methods to pass data directly to controls in the OPL cable condition monitoring process, to connect to HTTP VI methods, and to send values as POST data using the HTTP POST method. The URL that web clients use to exchange data with HTTP VI methods is based on several values that are determined when a LabVIEW web service is created. In a LabVIEW cluster, each element can have a name, making it easy to modify clusters using the unbundle/bundle by name functions [24].

The application program for the OPL cable health monitoring process uses URLs and HTTP methods to pass real-time data to the controls in the connector panel based on the Web VI method, and to pass values with the post-data role using the HTTP POST method. The HyperText Markup Language (HTML) code allows for all the visual aspects of the Web page and is similar to the front panel, which contains indicators and graphs to display the results. It also uses JavaScript code for this purpose. Communication between the LabVIEW application and JavaScript is through a LabVIEW Web Service Request, which is a service in the LabVIEW environment. This allows for the creation, writing and reading of global variables to achieve the above objectives, as shown in Figure 21.

This OPL cable condition monitoring software application will use a function that converts LabVIEW clusters to JSON objects. The block diagram of the communication modules between the application program and the Web server integrated into LabVIEW is shown in Figure 22a. Figure 22b shows the graphical interface available to a user from the electricity provider’s intranet to view the parameters provided by the OPL cable condition monitoring software application. The following URL is used to call the web client software application: “http://127.0.0.1:8001/webcablesag/cablemonitoring.html” accessed on 14 May 2024.

The configuration of the monitored parameter fields of the OPL cables on a secure channel from the ThingSpeak IoT cloud platform is shown in Figure 23. It can be seen that on the channel created with the ID 2284751, the fields for experimental data have been configured as follows: cable left tilt [°], cable right tilt [°], cable mechanical stress *H* [kN], and (*x**, *y**) cable sag values point.

The platform can be accessed from the LabVIEW programming environment using GET, POST, PUT or DELETE context commands. These commands represent HTTP requests and responses, and are part of the Representational State Transfer (REST) architecture; they are presented for this application in Figure 24. The block diagram of the LabVIEW client software module for writing OPL cable monitoring process data to the ThingSpeak IoT cloud platform is shown in Figure 24a, and the IoT LabVIEW client software interface module is shown in Figure 24b. In the interface in Figure 24b, the data obtained from the experiments are retrieved via global variables in the project structure from the LabVIEW software module application client for reading values from the sensor network via a ModBUS I/O server and Matlab and LabVIEW software module application for real-time calculation of OPL cable sag described in Section 4.

Viewing the data recorded in the form of charts on the ThingSpeak IoT cloud platform with real-time data provided by the software application interface for the OPL cable monitoring process can be accessed using a user ID and password at the address “https://thingspeak.com/channels/22884751”, as shown in Figure 25.

The user can view the results of the application in real real-time in the ThingSpeak IoT cloud platform, but can also save a set of data on demand, which can be processed and analysed offline, as shown in Figure 26. For example, the following quantities with experimental data are shown: tilt 1 in Figure 26a, tilt 2 in Figure 26b, mechanical cable stress in Figure 26c, Cable weight/length in Figure 26d, cable sag *x** in Figure 26e, and cable sag *y** in Figure 26f.

### 5.2. Discussion

OPL monitoring is an essential component of power grid operations. An effective monitoring system can help identify and fix potential problems before they cause outages or failures. Measuring the performance of the OPL monitoring system can be achieved by collecting and analysing data from sensors such as currents, mechanical condition of the line and temperatures. The data can be used to identify trends, anomalies and other potential problems.

The presented monitoring system provides a highly accurate database, which supports the safe operation of OPL at the maximum permissible load. Also, the presented monitoring system allows the monitoring and diagnostic structures to be optimized in terms of efficiency and cost. The overall aim of the optimisation is to achieve the following objectives: monitoring and identification of the condition of OPL; loading related to the condition of OPL, identification of sub-assemblies with vulnerabilities and location of evolving faults; identification of critical wear thresholds and the dynamics of fault evolution; identification of causes generating excessive stresses; carrying out maintenance processes and durations between major interventions resulting in temporary equipment shutdowns; estimation of equipment lifetime for efficient maintenance.

For example, for the measurement of OPL cables sag in at-the-edge conditions, as shown in Figure 16, using real-time calculation software, the sag value (*x** = 14.07 m and *y** = −1.76 m) is obtained as in Figure 21. Using the offline software presented in Section 2, for the same conditions, the value of the sag (*x** = 14.06 m and *y** = −1.75 m) is obtained. Also, a physical measurement using a theodolite measuring system can obtain the exact value of the cable sag, which in this case is (*x** = 14.062 m and *y** = −1.758 m). A very small error is observed in the use of offline and real-time software in the calculation of the cable sag, compared to a value considered as standard obtained by other means. The difference between the values provided by the offline software and the real-time software is due to the rounding of the values transmitted during the communication between the sensors and the real-time system. In any case, the differences are negligible compared to the overall requirements imposed on the system. Since under laboratory conditions, real operating conditions (wind, rain, ice and temperature differences over a wide range) can only be simulated for very short periods, in real operation to improve the measurement accuracy of the system over time, the necessary corrections can be made using machine learning based software.

The presented system allows continuous monitoring of the value of OPL cable sag (on one or more sections of the line), and the data provided allows appropriate maintenance decisions to be made and contributes to the optimization of network operation through the possibility of dynamic charging as opposed to the traditional static charging. Through the hardware and software architecture of the OPL condition monitoring system including the calculation of cable sag, but also the use of high accuracy, precision and reliability hardware components, it can be said that the presented monitoring system is robust, reliable and easily allows the scalability of the software application in case more than one section is to be monitored. Thus, the calculations are performed with high accuracy due to the low latency of the sensor network, as it conveys a relatively small amount of data, and the overall period of providing calculations is of the order of seconds. Given the fact that most of the sensors have very good accuracy and precision provided by the datasheet (e.g., for the tilt sensor the accuracy is ±0.01°), and the maximum relative error accepted by the beneficiary (the power electricity transporter company) is ±0.05 m, it can be said that the measurement and calculation system offers very good accuracy and precision.

## 6. Conclusions

This article presents the implementation of a hardware and software system for online monitoring of OPL electricity transport parameters and the determination of OPL cable sag. The mathematical model based on differential equations together with the algorithm for calculating the cable sag are presented. Given that, based on the mathematical model presented, the calculation of the OPL cable sag can be done in different ways depending on the sensors used. The presented application uses a variety of sensors, and a direct calculation is made through a method that follows the highlighted verification relationships directly, and the calculation is through an alternative method that uses another set of sensors to generate both verification of the calculation and the functionality of the sensors, thus obtaining a sensor fault observer.

The hardware architecture of the OPL cable online monitoring application is presented together with the main characteristics of the sensors and communication equipment used. The configurations required to transmit data using the ModBUS and ZigBee protocols are also presented. Ontext are the main software modules of the OPL cable condition monitoring process application, which ensure the monitoring of the main parameters of the power line and the visualisation of the results both on the electricity provider’s intranet using a web server and a MySQL database, and on the Internet using an IoT server. This categorisation of data visualisation mode is done to ensure a high level of cybersecurity. The OPL cable sag calculation algorithms are presented in both simulated and real time. The development software environments used are Matlab and LabVIEW. Also, the global accuracy of the entire OPL cable sag calculus system is estimated at 0.1%. The main modules of the application software are the following: the software module for reading quantities from the sensor network via a ModBUS I/O server, the software module for real-time calculation of the OPL cable sag, the software module for reading/writing from a MySQL type database, the software module client application for sending automatic e-mails with warnings and alarms regarding the limits of the OPL cable sag and sensor faults, the software module for data transmission on the wholesale electricity provider’s intranet via a web server, and the software module for viewing data by authorised users based on user and password via an IoT server.

In future work, as a result of collaboration with users, data will be collected both from the presented application and from direct measurements in extreme climatic conditions (wind and ice) to make the necessary corrections using software based on machine learning. This is because the mathematical model cannot accurately reproduce the sag determination of OPL cables in these contexts.

## Figures and Tables

**Figure 1 sensors-24-04283-f001:**
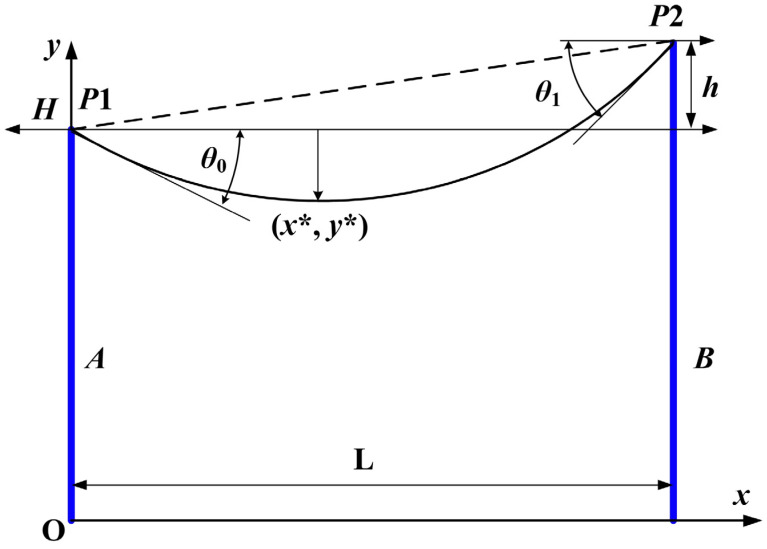
Geometric representation of the cable sag.

**Figure 2 sensors-24-04283-f002:**
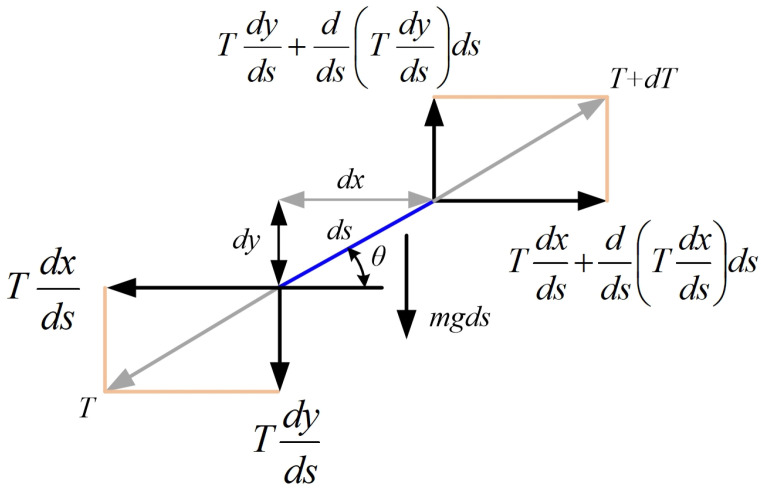
Equilibrium of an infinitesimal element of the inextensible cable under gravitational load.

**Figure 3 sensors-24-04283-f003:**
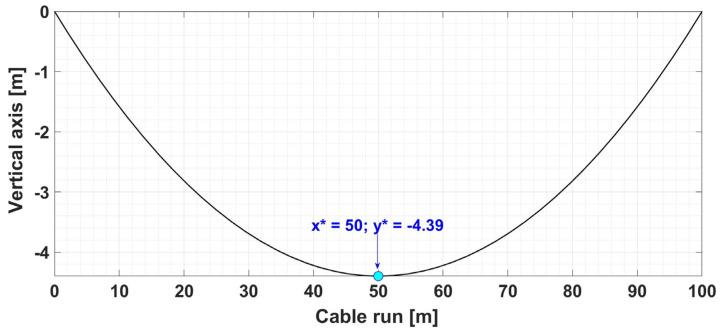
Numerical simulation and Matlab graphical representation of the cable sag calculus for example 1.

**Figure 4 sensors-24-04283-f004:**
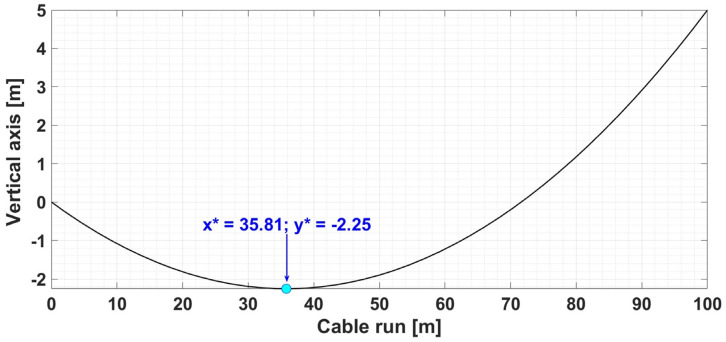
Numerical simulation and Matlab graphical representation of the cable sag calculus for example 2.

**Figure 5 sensors-24-04283-f005:**
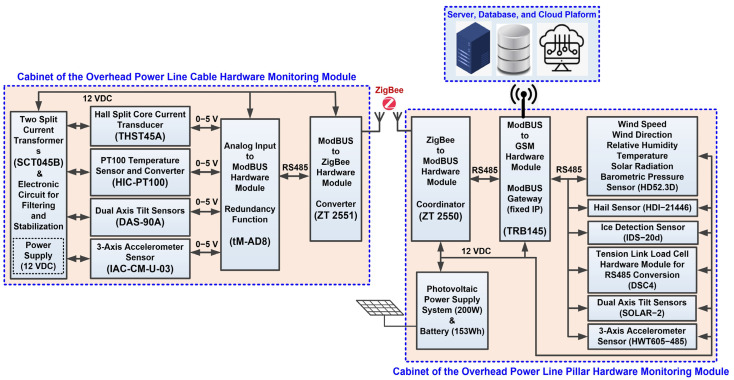
Proposed general hardware architecture of the OPL cable monitoring system.

**Figure 6 sensors-24-04283-f006:**
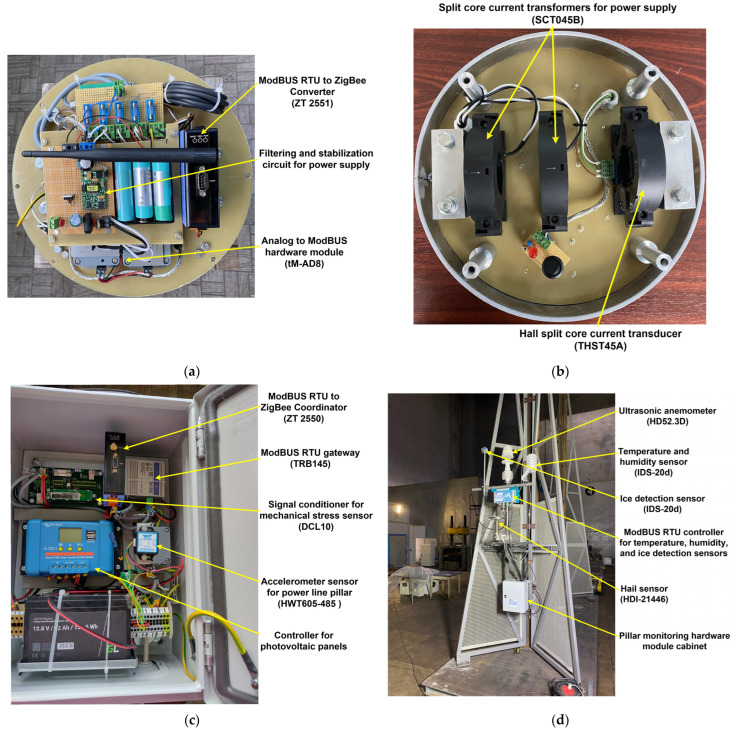
Sensor network for the OPL cable monitoring system: (**a**) Board 1 of the cabinet from the OPL cables with the sensors for measuring the state of the cable; (**b**) Board 2 of the cabinet from the OPL cables with split core current transformers and Hall split-core current transducer; (**c**) Cabinet of the pillar with hardware modules; (**d**) Cabinet of the pillar with the climatic conditions sensors.

**Figure 7 sensors-24-04283-f007:**
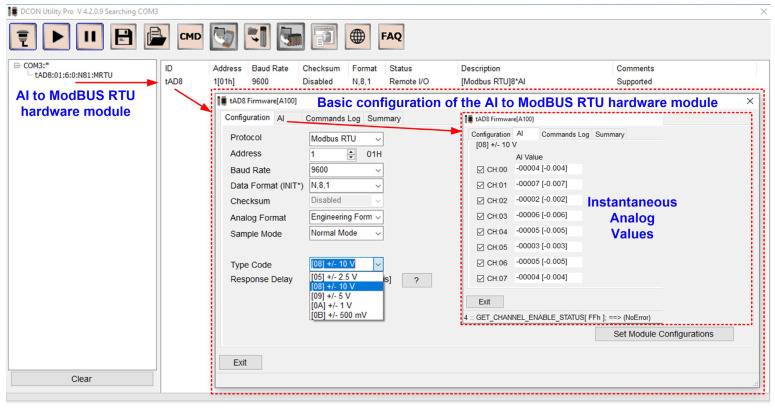
Analogue input to ModBUS RTU hardware module basic configuration.

**Figure 8 sensors-24-04283-f008:**
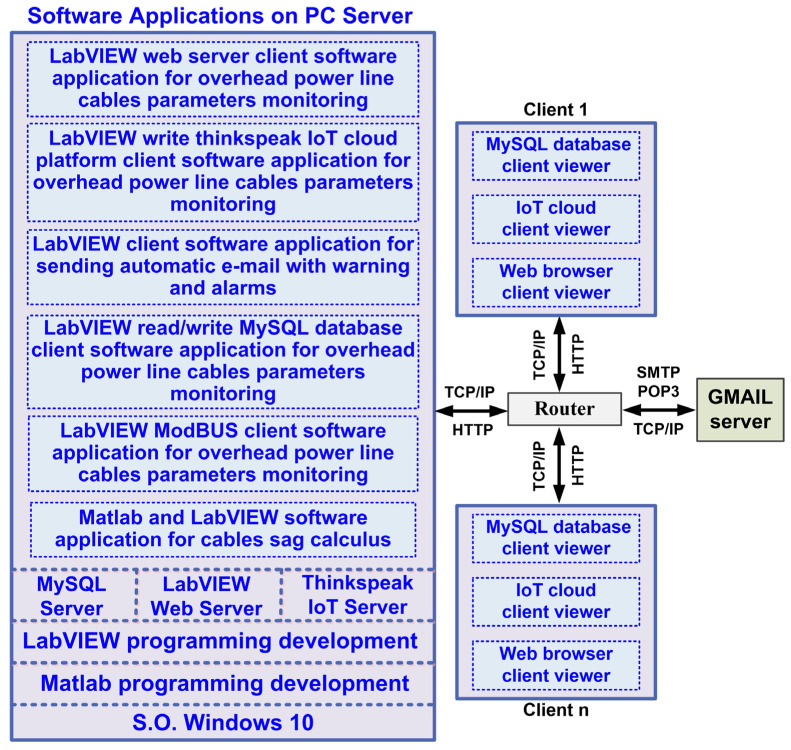
Software architecture of the OPL cable monitoring system.

**Figure 9 sensors-24-04283-f009:**
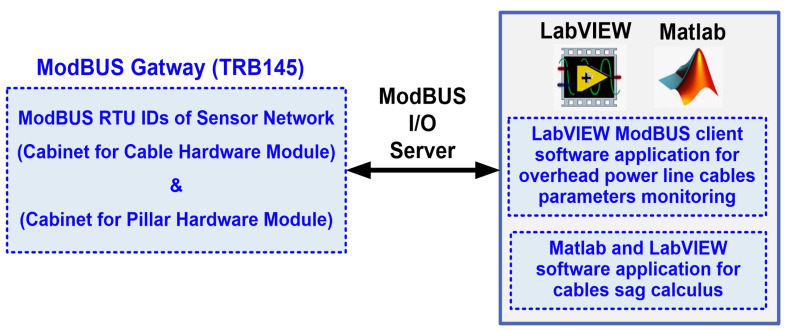
LabVIEW project configuration of the ModBUS I/O Server for reading real-time data from the sensor network.

**Figure 10 sensors-24-04283-f010:**
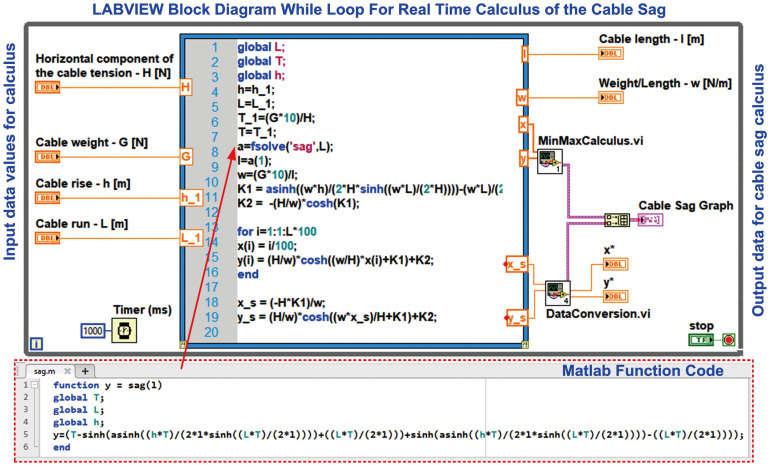
LabVIEW and Matlab software block diagram for real-time calculation of the cable sag value.

**Figure 11 sensors-24-04283-f011:**
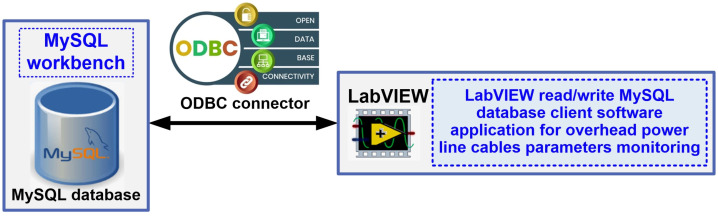
MySQL database and LabVIEW connection configuration using the ODBC connector.

**Figure 12 sensors-24-04283-f012:**
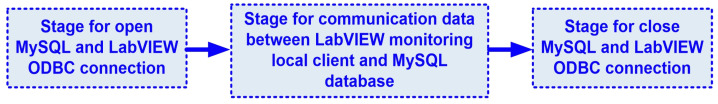
Block diagram for writing data from the OPL cable monitoring process to database (LabVIEW client to MySQL database).

**Figure 13 sensors-24-04283-f013:**
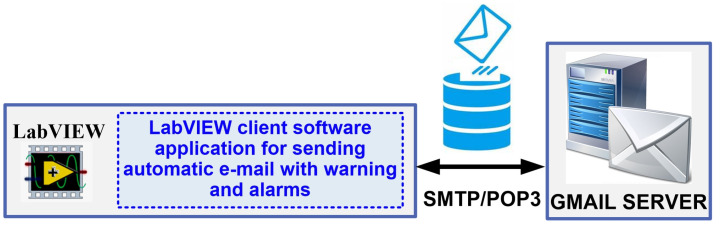
Block diagram of the automated sending warning email from OPL cable monitoring.

**Figure 14 sensors-24-04283-f014:**
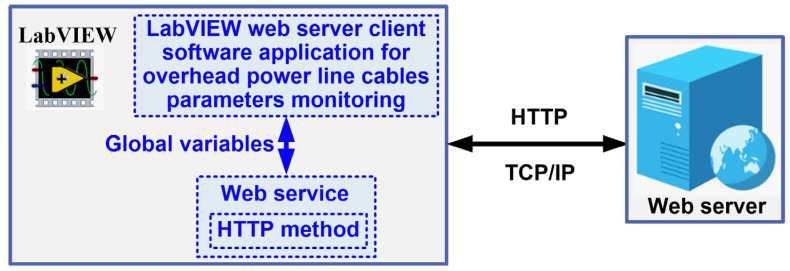
Block diagram of the web server application with data from OPL cable monitoring.

**Figure 15 sensors-24-04283-f015:**
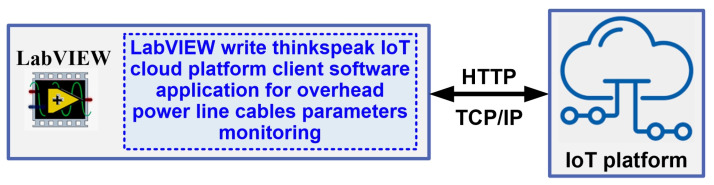
Block diagram of IoT cloud platform with data from OPL cable monitoring.

**Figure 16 sensors-24-04283-f016:**
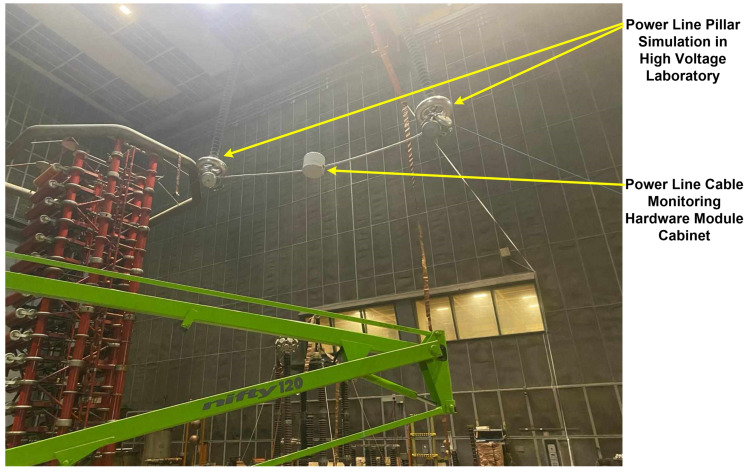
OPL cables monitoring system testing in laboratory conditions.

**Figure 17 sensors-24-04283-f017:**
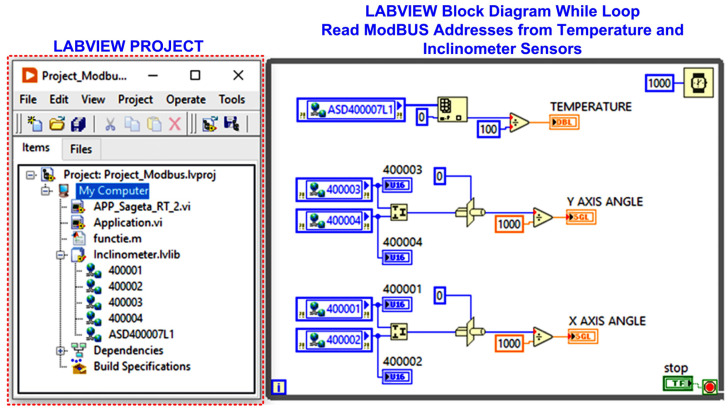
LabVIEW project and block diagram for real-time data acquisition from tilt sensors via ModBUS I/O Server.

**Figure 18 sensors-24-04283-f018:**
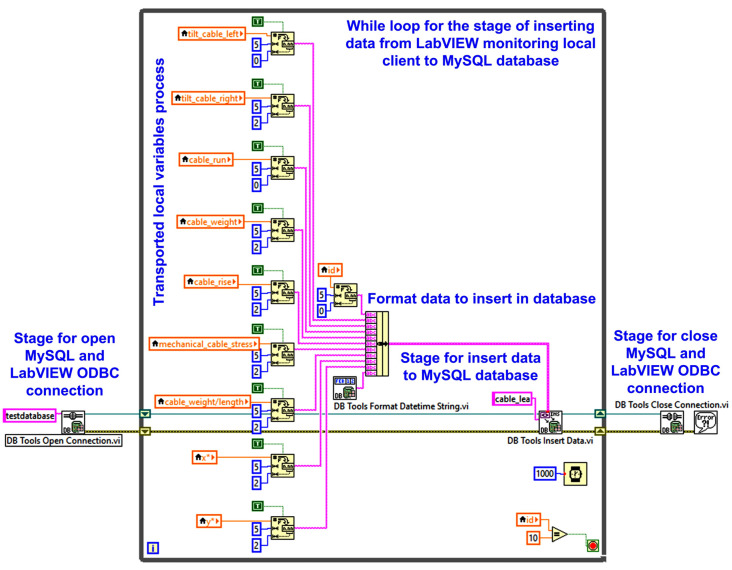
LabVIEW client software application block diagram for writing cable monitoring data to the MySQL database.

**Figure 19 sensors-24-04283-f019:**
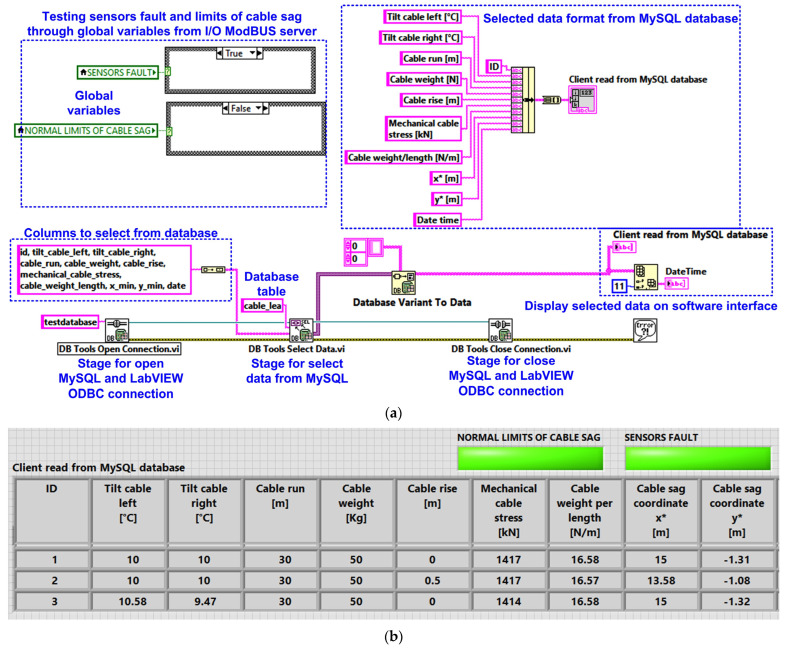
LabVIEW client software application for reading cable monitoring data from the MySQL database: (**a**) LabVIEW software block diagram; (**b**) LabVIEW software interface.

**Figure 20 sensors-24-04283-f020:**
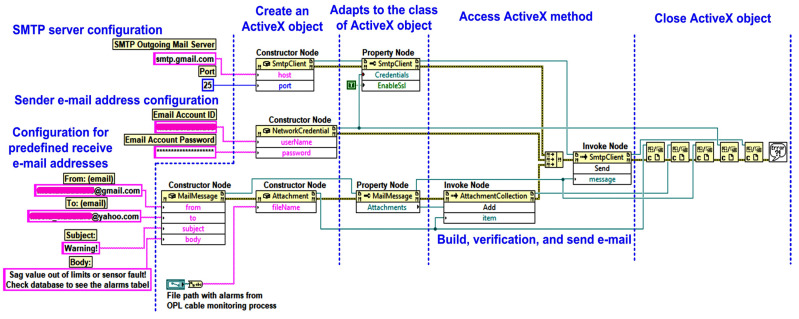
LabVIEW software block diagram of automated sending email process.

**Figure 21 sensors-24-04283-f021:**
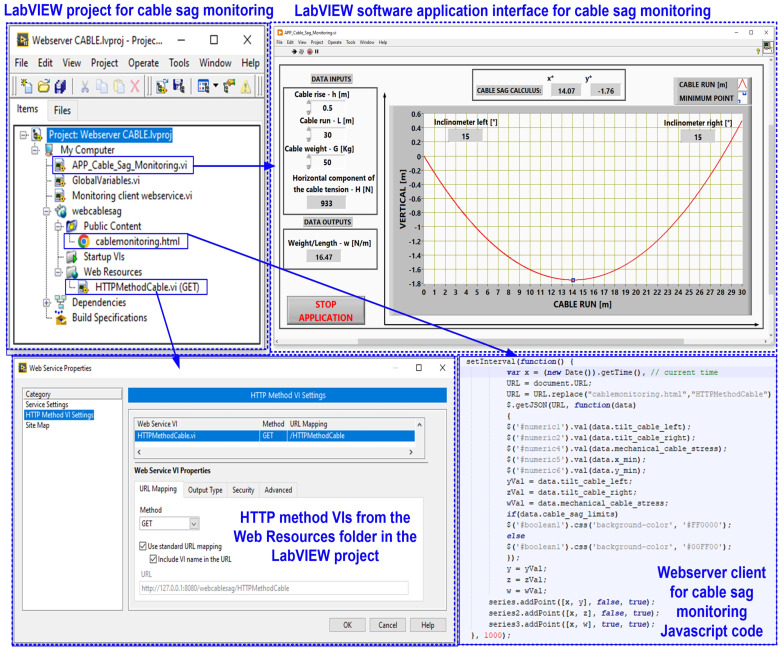
LabVIEW web server project of real-time software application for OPL cable monitoring process.

**Figure 22 sensors-24-04283-f022:**
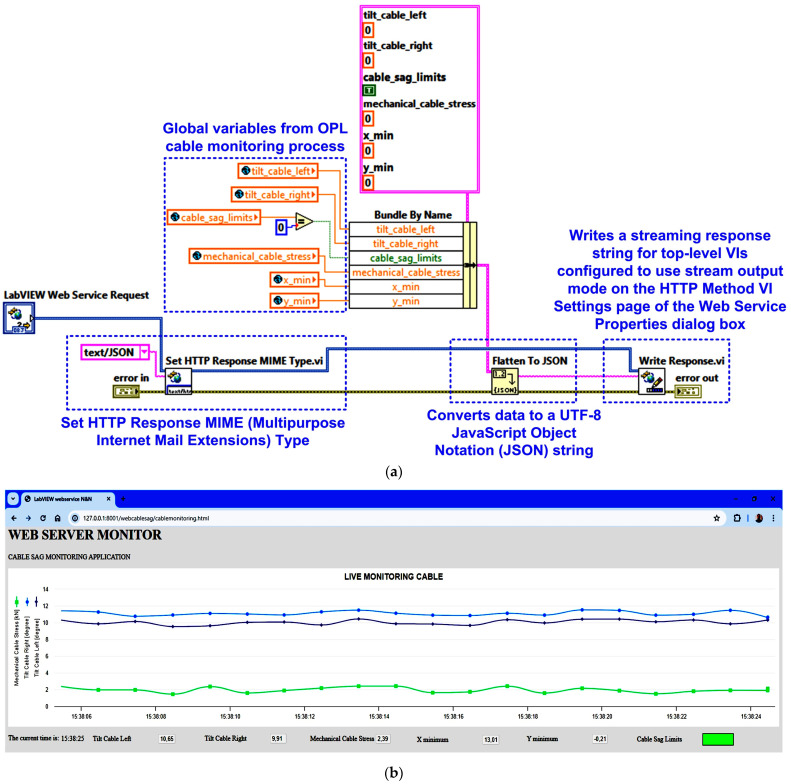
LabVIEW client software application for reading the cable monitoring process data from MySQL database: (**a**) LabVIEW software block diagram of the HTTP method; (**b**) client browser software interface.

**Figure 23 sensors-24-04283-f023:**
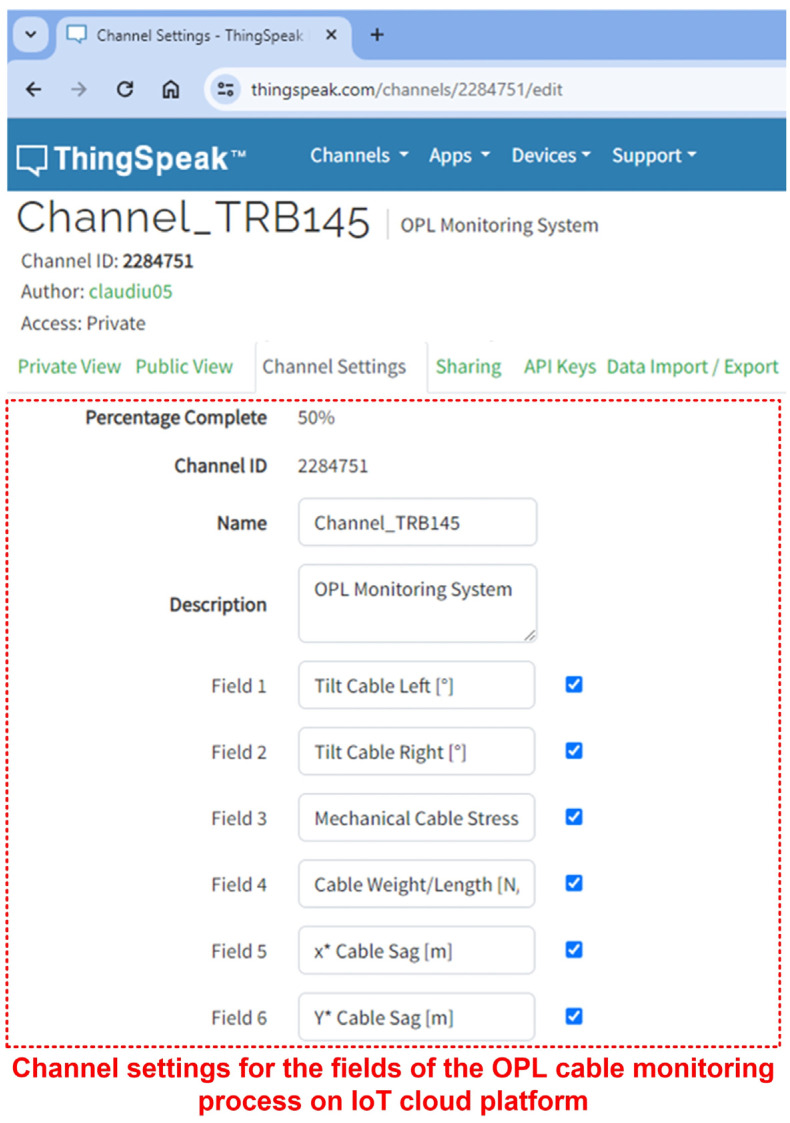
ThingSpeak IoT cloud platform with data from OPL cable monitoring process—channel settings for experimental data fields.

**Figure 24 sensors-24-04283-f024:**
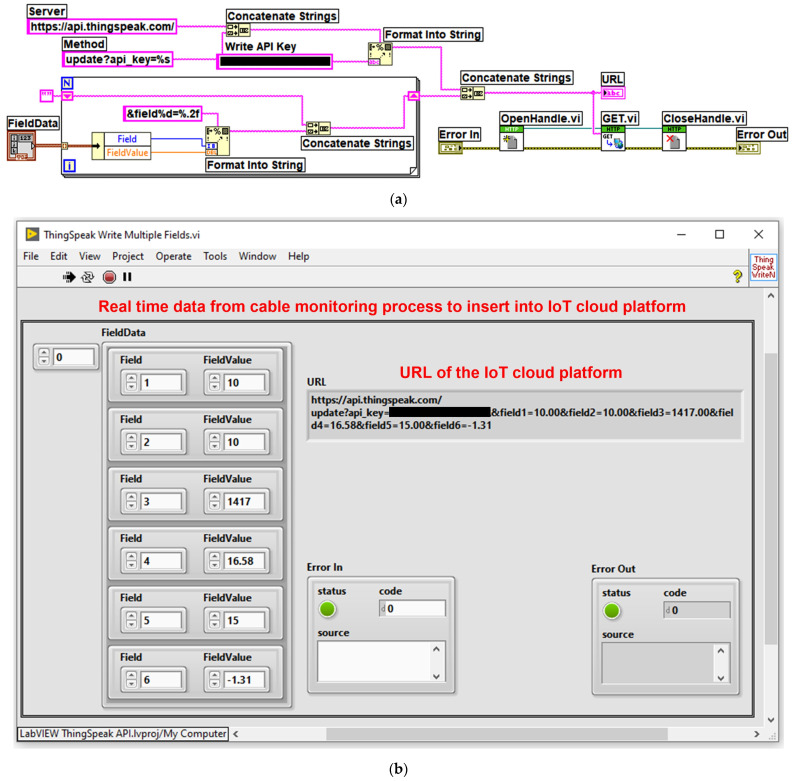
LabVIEW client software application for writing cable monitoring process data to IoT cloud platform: (**a**) LabVIEW software block diagram for sending data; (**b**) IoT LabVIEW client software interface.

**Figure 25 sensors-24-04283-f025:**
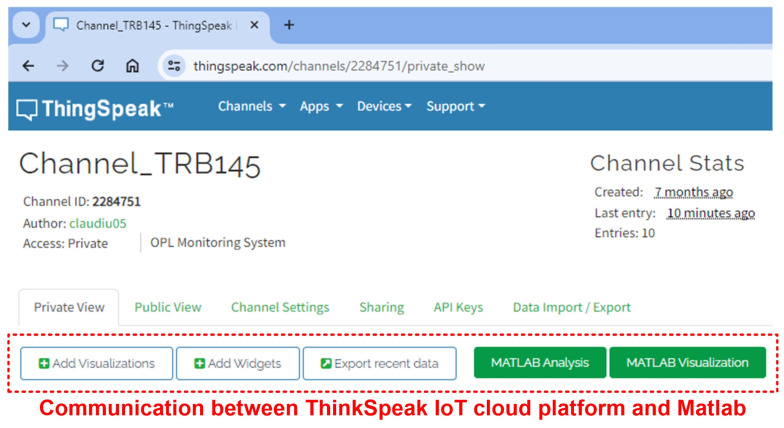
ThingSpeak IoT cloud platform interface for OPL cable monitoring process.

**Figure 26 sensors-24-04283-f026:**
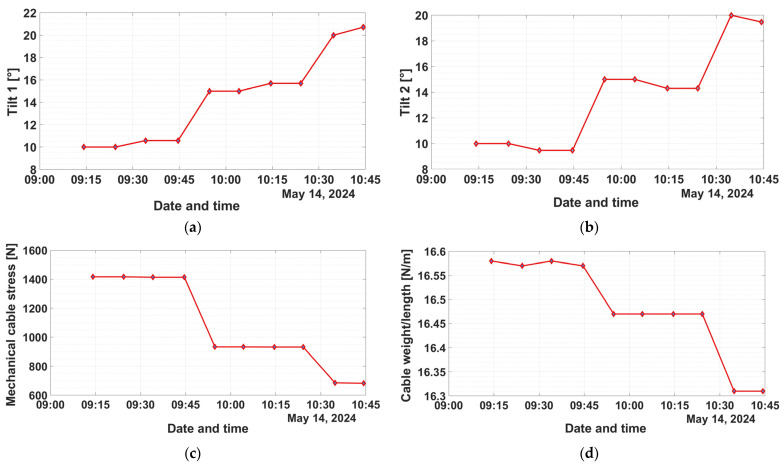
OPL cable monitoring real-time data from ThingSpeak IoT cloud platform—Matlab charts visualization: (**a**) Tilt 1; (**b**) Tilt 2; (**c**) Mechanical cable stress; (**d**) Cable weight/length; (**e**) Cable sag *x**; (**f**) Cable sag *y**.

**Table 1 sensors-24-04283-t001:** OPL cable sag measurement methods.

Sag Measurement Method	Features
Detection method	This type of method is simple. It has a high degree of accuracy and the need for on-site operators precludes the possibility of real-time monitoring.
GPS	This type of method is accurate, but involves complex algorithms that are difficult to implement and costly.
Temperature and cable tension	This type of method is relatively inexpensive, it involves the measurement of too many parameters, the calculation algorithm is complex, and it introduces large errors.
Projection technique	This type of method is easy to implement, it requires specialised photographers, image processing is difficult, and it introduces large errors.
Line slope	This type of method is relatively simple and inexpensive, the calculation algorithm is simple, and it introduces small errors.

**Table 2 sensors-24-04283-t002:** Input data and calculated data using cable sag calculation algorithm—Example 1.

Input Data				Output Data		
Cable Run *L* [m]	Cable Weight *G* [N]	Cable Rise *h* [m]	Mechanical Cable Stress *H* [N]	Cable Weight/Length *w* [N/m]	*x**	*y**
100	150	0	4235	14.9	50	−4.39

**Table 3 sensors-24-04283-t003:** Input data and calculated data using cable sag calculation algorithm—Example 2.

Input Data				Output Data		
Cable Run *L* [m]	Cable Weight *G* [N]	Cable Rise *h* [m]	Mechanical Cable Stress *H* [N]	Cable Weight/Length *w* [N/m]	*x**	*y**
100	150	0.5	4235	14.90	35.81	−2.25

**Table 4 sensors-24-04283-t004:** Recorded data in MySQL database workbench from OPL cable monitoring process.

Real-Time Input Data				Calculated Output Data		
Cable Tilt 1 [°]	Cable Tilt 2 [°]	Cable Rise *h* [m]	Mechanical Cable Stress *H* [N]	Cable Weight/Length *w* [N/m]	*x** [m]	*y** [m]
10	10	0	1417	16.58	15	−1.31
10	10	0.5	1417	16.57	13.58	−1.08
10.58	9.47	0	1414	16.58	15	−1.32
10.58	9.47	0.5	1414	16.57	13.58	−1.08
15	15	0	933	16.47	15	−1.99
15	15	0.5	933	16.47	14.06	−1.75
15.7	14.3	0	932	16.47	15	−1.99
15.7	14.3	0.5	932	16.47	14.06	−1.75
20	20	0	686	16.31	15	−2.7
20.72	19.47	0.5	683	16.31	15	−2.71

## Data Availability

Data are contained within the article.

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
