# Peer review of "Real-Time Monitoring of Cable Sag and Overhead Power Line Parameters Based on a Distributed Sensor Network and Implementation in a Web Server and IoT"

_sensors, 2024, doi:10.3390/s24134283_

Round 1

Reviewer 1 Report (New Reviewer)

Comments and Suggestions for Authors

Dear Authors, please find my review about the paper, titled "Real-time Monitoring of Cable Sag and Overhead Line Parameters Based on a Distributed Sensor Network and Implementation in a Web Server and IoT" below.

1. Abstract is too long, please compress it without losing meaning. The whole paper is a bit overwritten and bloated with figures, without emphasizing scientific merit. It could be reduced a bit in length. Some figures are also showing more than it is required in a paper. E.g. Figure 9 is a bit unneccessary for understanding. Also figures 10, 11, 12, 13, 15, 18, 20, 22, 26, 27 are screenshots, which are not proper for this kind of publication. I'd remove them and highlight in text, what is necessary. (They could be valuable in a white paper, for example.)
Please export the actual data plot from Figure 25, there is no need to put image"junk" with browser headers, software windows, etc. Same for Figure 28. Export figures as own charts, not a Thingspeak browser screenshot.

2. Some figures use serif-based fonts, which are more difficult to read in technical documentations. Figure 7 is one good example, where furthermore, i'd detach the descriptive textboxes from the photos, and put them outside, even if the figures will take more space. Same for Figure 24. Fig 14 is also too dense with serif-based (and CAPTIAL) letters. I'd redraw with basic Arial (or similar) font types.

3. I'd recommend to copy and paste the calculus.m as text into the body of the manuscript, and use a styling, accurate for code.

4. Discussion is weak on the gathered data. I would like to see a quantiative and qualitative analyis on the recorded data, to amplify scientific quality of the paper. This could be a real added value of the work.

5. I'd recommend adding a nomenclature and a list of abbreviations to the paper.

Comments on the Quality of English Language

OK

Author Response

Dear reviewer, thanks for your recommendations.

  1. We have made the requested changes to the Abstract of this article. We have removed some of the figures, others figures we replaced them with ones built from scratch and inserted the corresponding explanations in the text. The rest of the mentioned figures have been kept because they are, in our opinion, connecting elements in the logical chain of the steps of the software application.
  2. We have made the requested changes. We have redone the figures in question.
  3. We have made the requested changes. Thus, we have made in the article the presentation of "calculus.m" in the form of Algorithm 1 according to the appropriate MDPI style.
  4. The presented system allows continuous monitoring of the value of overhead power line cable sags (on one or more sections of the line), and the data provided allows appropriate maintenance decisions to be made, and contributes to the optimization of network operation through the possibility of dynamic charging as opposed to the traditional static charging. Through the hardware and software architecture of the overhead power line condition monitoring system including the calculation of cable sag, but also through the use of high accuracy, precision and reliability hardware components, it can be said that the presented monitoring system is robust, reliable and easily allows the scalability of the software application in case more than one section is to be monitored. Within the calculation algorithm, corrections can be introduced in the form of equation (22) which allows corrections with temperature of the measurements taken from the sensors and implicitly corrections of the calculated value. Thus, the calculations are performed with high accuracy and due to the low latency of the sensor network as it conveys a relatively small amount of data, and the overall period of providing calculations is of the order of seconds. Given the fact that most of the sensors have very good accuracy and precision provided by the data sheet (e.g. for the tilt sensor the accuracy is ± 0.01°), and the maximum relative error accepted by the beneficiary (the electricity transporter) is ± 0.05m, it can be said that the measurement and calculation system as a whole offers very good accuracy and precision. As far as the testing of the system is concerned, this was done under quasi-real conditions in our own laboratory (High Voltage Laboratory), where the maximum distance between two pillars is less than 40 m and the height up to 30 m. In future work, as a result of collaboration with users, data will be collected both from the presented application and from direct measurements in extreme climatic conditions (wind and ice) in order to make the necessary corrections using software based on machine learning. This is because the mathematical model cannot accurately reproduce the sag determination of OPL cables in these contexts. A very small error is observed in the use of offline and real-time software in the calculation of the cable sag, compared to a value considered as standard obtained by other means. The difference between the values provided by the offline software and the real-time software is due to the rounding of the values transmitted during the communication between the sensors and the real-time system. In any case the differences are negligible compared to the overall requirements imposed on the system. Since under laboratory conditions, real operating conditions (wind, rain, ice and temperature differences over a wide range) can only be simulated for very short periods of time, in real operation to improve the measurement accuracy of the system over time, the necessary corrections can be made using machine learning based software.
  5. We have attached in reply to this point the requested nomenclature.

Nomenclature

OPL

Overhead Power Lines

SCADA

Supervisory Control and Data Acquisition

LabVIEW

Laboratory Virtual Instrument Engineering Workbench

VI

Virtual Instrument

TDMS

Technical Data Management Streaming

WPAN

Wireless Personal Area Network

PAN

Personal Area Networks

RTU

Remote Terminal Unit

LabVIEW DSC

LabVIEW Datalogging and Supervisory Control

OPC

Open Platform Communications

API

Application Programming Interface

ODBC

Open Database Connectivity

MySQL

Relational database management system developed by Oracle that is based on structured query language

SMTP

Simple Mail Transfer Protocol

IP

Internet Protocol

DDNS

Dynamic Domain Name System

GSM

Global System for Mobile Communications

GPRS

General Packet Radio Service

RTCM

Radio Technical Commission for Maritime Services

NTRIP

Networked Transport of RTCM via Internet Protocol

HTTP

Hypertext Transfer Protocol

URL

Uniform Resource Locator

JSON

JavaScript Object Notation

HTML

HyperText Markup Language

REST

Representational State Transfer

IoT

Internet of Things

Reviewer 2 Report (New Reviewer)

Comments and Suggestions for Authors

This paper presented the implementation of a hardware and software system for on-line monitoring of OPL electricity transport parameters and determination of OPL cable sag. The mathematical model based on differential equations and the method of algorithmic calculation of OPL cables sag are presented.. Simulation results are given. The paper is generally well written. However, there are also some issues as listed below.

1)      Simplify and clarify the abstract to make it more accessible. Avoid excessive technical jargon and ensure that the problem, method, and results are concisely summarized.

2)      Expand the introduction to better contextualize the importance of monitoring overhead power lines. Include more background information and explain why current methods are insufficient and enhance the literature review by including more recent studies and comparing the proposed method with existing ones in greater detail. Highlight the novel contributions of your work more explicitly.

3)      For Hardware Architecture, please provide a more detailed diagram of the hardware architecture. Label each component clearly and describe its function within the system.

4)      For  Communication Protocols, could authors elaborate on the choice of ModBUS and ZigBee protocols. Discuss their advantages and limitations in the context of this application and compare them with other potential protocols.

5)      Please improve the presentation of results by including more graphs, tables, and comparative analyses. Show how the system performs under different conditions and compare it with baseline methods.

6)      Please conduct a thorough error analysis to identify potential sources of error in the sensor data and calculation methods. Discuss how these errors were mitigated or accounted for.

Comments on the Quality of English Language

Extensive editing of English language required.

Author Response

Dear reviewer, thanks for your recommendations.

  1. We have made the requested changes to the Abstract of this article.
  2. In Table 1 some methods of determining cable sag are presented highlighting the advantages and disadvantages of the method. In the presented article we point out that the determination of cable sags is done in a redundant way, since from the point of view of the calculation algorithm one can use the inclination angles at the ends of the cable and respectively one of the angles and the mechanical cable stress in the cable. The overhead power line monitoring system can also include several such sections where cable sag is monitored, along with a range of signals provided by specialised sensors on weather conditions, cable vibration, pillar vibration and overhead power line cable current. The logical chain of the software elements is based on the Matlab environment for the numerical simulations, while for the real-time implementation the LabVIEW development environment is used, which is suitable for the global integration of the system including Matlab scripts for calculations, but also the integration of a MySQL database, a web server and an IoT server. Thus, the main contributions of this article are as follows:
    • Establishing alternative calculation methods and verification calculation relations for the OPL cable sag calculation depending on the groups of sensors used, thus obtaining a software method for detecting the failure of a sensor, a method whose usefulness can be expressed in terms of increasing system reliability;
    • Establishing the hardware and software architecture of the OPL online cable monitoring application;
    • Setting up the necessary configurations for data transmission using ModBUS and ZigBee protocols in order to minimise energy consumption in the process of local centralisation of data from sensors and transmission via Global System for Mobile Communications (GSM) to the user server;
    • Implementation of the software module for reading quantities from the sensor network through a ModBUS I/O server;
    • Implementation of the software module for the real-time calculation of the OPL cable sag and monitoring of the parameters of interest of the power line, including the correct functioning of the sensors and maintenance within the acceptance limits of the OPL cable sag values;
    • Implementation of a software module to send automatic e-mails with warnings and alarms regarding OPL cable sag limits and sensor faults;
    • Implementation of the software module for the transfer of data to the user's Internet via a web server;
    • Implementation of the software module for viewing data by authorised users, based on user and password, via an IoT server.
  3. We have made the requested changes. Thus, elements of the hardware architecture have been labelled and details of these devices have been presented in the text. The presentation was made by a suitable grouping on the cabinets made placed on the overhead power line cable respectively on the pillar.
  4. Given the particularities of the application presented (power supply possibilities, spatial location and number of variables that form the data traffic), the choice of ModBUS and ZigBee protocols is optimal. Among the advantages of these protocols we mention: the ModBUS protocol is a communication protocol based on a master-slave or client-server architecture, and the main purpose of the ModBUS protocol is to facilitate stable and fast communication between the slave hardware subsystem in the ZigBee wireless network and the sensors in the subsystem component OPL cable monitoring hardware. ZigBee, as described in the IEEE 802.15.4 standard, is a low bit rate data transfer protocol for Wireless Personal Area Networks (WPAN). It is designed for easy connection between devices, keeping power consumption to an optimum minimum. The ZigBee network is self-organising, requiring minimal user or administrator intervention during initial configuration. Further intervention is only required in major problem situations where a very large number of nodes fail, or where running configurations are deleted and reset.
  5. Figures 15 and 16 have been inserted to present data from laboratory experiments for the real-time overhead power line cable monitoring system. Also, the facilities on the evolution of cable sag calculation is presented in Figures 20 and 23. For example, for the measurement of cables sag in at-the-edge conditions, as shown in Figure 19, using real-time calculation software, the sag value (x* = 14.07m and y* = -76m) is obtained as in Figure 18. Using the offline software presented in Section 2, for the same conditions the value of the sag (x* = 14.06m and y* = -1.75m) is obtained. Also, a physical measurement using a theodolite measuring system can obtain the exact value of the cable sag which in this case is (x* = 14.062m and y* = -1.758m). A very small error is observed in the use of offline and real-time software in the calculation of the cable sag, compared to a value considered as standard obtained by other means. The difference between the values provided by the offline software and the real-time software is due to the rounding of the values transmitted during the communication between the sensors and the real-time system. In any case the differences are negligible compared to the overall requirements imposed on the system. Since under laboratory conditions, real operating conditions (wind, rain, ice and temperature differences over a wide range) can only be simulated for very short periods of time, in real operation to improve the measurement accuracy of the system over time, the necessary corrections can be made using machine learning based software.
  6. Following the calculation procedure described in Algorithm 1, a relatively small amount of mathematical processing is observed for the determination of the cable sag, by processing the quantities of the geometrical characteristics specific to the line (distance between pillars, height difference, specific cable weight) and two quantities acquired from the inclination sensors and from the sensor providing the mechanical cable stress in the cable. Given the fact that most of the sensors have very good accuracy and precision provided by the catalogue sheet (for example for the tilt sensor the accuracy is ± 0.01°), and the maximum relative error accepted by the beneficiary (the electricity transporter) is ± 0.05m it can be said that the measurement and calculation system as a whole offers very good accuracy and precision. Also in relation (22), a temperature correction term for the magnitude correction provided by the tilt sensor is shown, such corrections being provided by the catalog sheets of all sensors used. Thus, we can say that the calculations are performed with high accuracy and due to the low latency of the sensor network since it conveys a relatively small amount of data, and the overall period of providing calculations is of the order of seconds.

Round 2

Reviewer 1 Report (New Reviewer)

Comments and Suggestions for Authors

Q1: Thank you for the changes introduced. There are some that were not taken seriously.

Q2: Text-box based figures should be saved and imported vectorgraphically. (SVG, EMF, WMF, etc...) Still, I'm not too fond of the serif-based fonts, and the texting in the legend boxes (eg. in Figure 6, where the text looks too dense, and at points, the words are broken by the end of the lines.) This is outlook, but decreases overall quality. Same for Fig 9. Please CTRL+A on the figures and change fonts.

Q3: VERY IMPORTANT! Qualitative data should be added to abstract and conclusions too. (There was little change in this.) Without this, the paper is an engineering report, not an "original research paper".

Q4: Let's get to the "results" figures then.
- Fig 10 - is an image on how to connect 3 axes. There is no added value, albeit it can have some point in the "experimental" section of a paper.
- Fig 11 - is interesting, but only because of the "m" insertion. The right side of the image is completely undecipherable on its own.
- Fig 12 - OK
- Fig 13 - could be done in block diagrams without the LabView schematic, which is meaningless in a journal paper. The image is suitable for the "experimental" section of a paper.
- Fig 14 - completely unnecessary; no one will get any added value from it. It is a journal paper, not a dev documentation.
- Fig 15a - The data would be interesting, but the paper shows no interest in the actual calculations, only to show what the VI looks like.
- Fig 15b  is interesting, but could be done without the LabView schematic. I suggest a basic block diagram. The image is suitable for "experimental" section of a paper.
- Fig 16 - is interesting, but could be done without the LabView schematic. I suggest a basic block diagram. The image is suitable for "experimental" section of a paper. The showing of the email is also problematic. Parts of the figure are undecipherable.
- Fig 17 - is again for showing and documenting. The image is suitable for "experimental" section of a paper.
- Fig 18a - is interesting, but could be done without the LabView schematic. I suggest a basic block diagram. The image is suitable for "experimental" section of a paper.
- Fig 18b - real data from live monitoring. Would be interesting, but not as a screenshot from a browser but as exported, and reconstructed image (Origin, Matlab, Excel... as the basic requirements of a journal paper). What is there to be seen on the results? Are there any findings?
- Fig 20. Completely unneccessary screenshot. There is no added value. Channel settings is not a result, but an experimental setting, which is not really important from the aspect of a journal publication.
Fig 21. - Same as above.
- Fig 22 - real data from live monitoring. Would be interesting, but not as a screenshot from a browser but as exported, and reconstructed image (Origin, Matlab, Excel... as the basic requirements of a journal paper). Thingspeak is basically Matlab, there is even a button on the top right corner for proper figure export.
What is there to be seen on the results? Are there any findings? What is the date? Days? Hours/mins?

Q5: Summing up, two figures deal with actual results; the rest are work documentation. This means most of the paper is "experimental description," not "results + discussion," as the journal would recommend.

The changes were not performed in a meaningful way in the previous round. If the authors could not come up with a focus on real results, not the documentation of their HWSW buildup, i'll recommend reject.

Author Response

Dear reviewer, thanks for your recommendations.

We have restructured the article taking into account your comments.

Round 3

Reviewer 1 Report (New Reviewer)

Comments and Suggestions for Authors

Paper was revised in depth, and i have to give that the authors put much effort to it.  The block diagrams, the discussion and the image extensions are super welcome.

Still i must recommend three minor points before publication.

1- please give QUANTITATIVE numerical values in abstract and conclusion, how the system is performing tasks with the mentioned (page 27 655) "very small error". Also please highlight the percentage of the error too to signify your work.

2- Figures now are much more understandable. But i might made a big stirrup with the sans-serif fonts and serif-based font debate on the images. Let me list the problem, as we have now mixed results.
Sans Serif: 6, 9, 10, 12, 13, 14, 15, 17, 18, 19, 20, 21, 22, 23, 25
Serif:  3, 4, 5, 7, 8, 11, 16, 24, 26
A harmonization must be done, as this is totally mixed now.

The following figures are still "distorted by ratio" - please correct.
Figure 7 (dcon utility screenshot image is distorted along Y)
Figure 22 (monitor live image is distorted along Y)

Author Response

Dear reviewer, thanks for your recommendations.

We reply to your comments below.

1- We have introduced in the abstract and conclusions the estimated value of the global error for the overhead power line cable sag calculation system.

2- In order to harmonize the fonts used in the figures of the article we have modified figures 3, 4, 5, 7, 8, 11, 16, 24, and 26 with Sans Serif fonts.

3- We have modified figures 2 and 22 to avoid distortion.

This manuscript is a resubmission of an earlier submission. The following is a list of the peer review reports and author responses from that submission.

Round 1

Reviewer 1 Report

Comments and Suggestions for Authors

Good research paper but needs some minor corrections:

1-The novelty of the paper must be clearly mentioned in the abstract and conclusion section.

2- More new references must be added in the literature review section.

3-Add a table in the introduction section to summarize the methods, advantages and disadvantages of the previous works.

4- A comprehensive Comparison with the previous works must be added in a table in the results section with its explanations.

5- Add a future work section in the revised manuscript.

Comments on the Quality of English Language

Minor editing of English language required

Author Response

Dear reviewer, thanks for your recommendations.

1- The introduction section presents the main contributions of this article. We believe that the first two points represent the essence of the novelty of this article:

  • Establishing alternative calculation methods and verification calculation relations for the OPL cable sag calculation depending on the groups of sensors used, thus obtaining a software method for detecting the failure of a sensor, a method whose usefulness can be expressed in terms of increasing system reliability;
  • Establishing the hardware and software architecture of the OPL online cable monitoring application.

The corresponding text is inserted in the abstract: “Considering that, based on the mathematical model presented, the calculation of OPL cable sag can be done in different ways depending on the sensors used, and the presented application uses a variety of sensors. Therefore, a direct calculation is made using one of the different methods. Subsequently, the verification relations are highlighted directly and, respectively, the calculation by the alternative method, which uses another group of sensors, to generate both a verification of the calculation and the functionality of the sensors, thus obtaining a defect observer of the sensors. The hardware architecture of the OPL cable online monitoring application is presented, together with the main characteristics of the sensors and communication equipment used.”

Also, in the conclusions section the corresponding text is inserted: “Given that, based on the mathematical model presented, the calculation of the OPL cable sag can be done in different ways depending on the sensors used, and the presented application uses a variety of sensors, a direct calculation is made through a method that follows the highlighted verification relationships directly, respectively the calculation through an alternative method that uses another set of sensors to generate both a verification of the calculation and the functionality of the sensors, thus obtaining a sensor fault observer. The hardware architecture of the OPL cable online monitoring application is presented, together with the main characteristics of the sensors and communication equipment used.”

2- In the references section 31 references are presented, most of which are from the last 5 years and cover both the methodology presented in this article and alternative methods presented in other articles.

3- In the paper presented, the line slope method has been used in principle, where based on the detection of the OPL cable slope angles, sag value can be obtained to which, in order to obtain an alternative verification relation, the measurement of the mechanical cable stress is also used, thus achieving increased accuracy and redundancy without significant burden on the hardware system. The table below summarises the methods, advantages and disadvantages of previous work. 

Sag measurement method

Features

Detection method

This type of method is simple. It has a high degree of accuracy and the need for on-site operating staff excludes the possibility of real-time monitoring.

GPS

This type of method is accurate, it involves complex algorithms which are difficult to implement, it is costly.

Temperature and cable tension

This type of method is relatively inexpensive, it involves the measurement of too many parameters, the calculation algorithm is complex and it introduces large errors.

Projection technique

This type of method is easy to implement, it requires specialized photographers, the processing of images is difficult and it introduces large errors.

Line slope

This type of method is relatively simple, inexpensive, the calculation algorithm is simple and it introduces small errors.

4- From an analytical point of view, starting from the differential equations, which are the basis of the sag measurement cathenary algorithm, in this paper two methods are combined in practice. The alternative methods Detection method, GPS and Projection technique, consist of direct measurements, which either cannot be considered in real time or introduce a high complexity to the algorithm, and a concrete comparison between the advantages and disadvantages of these methods are summarized in the table above. For a direct comparison between any of the methods you should actually implement all of them. This cannot be the subject of a single scientific article, so in this article the verification of the values obtained by the proposed algorithms is done in the laboratory experimental stage, by comparison with a direct measurement method (Detection method using a theodolite device for measurement).

5- In the final section the corresponding text is inserted: “In future work, as a result of collaboration with users, data will be collected both from the presented application and from direct measurements in extreme climatic conditions (wind and ice) in order to make the necessary corrections using software based on machine learning. This is because the mathematical model cannot accurately reproduce the sag determination of OPL cables in these contexts.”

Reviewer 2 Report

Comments and Suggestions for Authors

-       In the title it should be mentioned: mechanical parameter;

-       In the title it should be mentioned: “in laboratory”, not for real OPL!

-       The article presents an interesting application, under laboratory conditions, but which faces major difficulties in practical implementation (on OPL, in real condition);

-       The work has a more theoretical character. How reliable would OPL be if so many sensors were installed between every two electricity pillars?

-       Pp.1-3, Section 1: OK;

-       P.4, Eq.1, No.3: Replace “sin” with “tan”, Verify carefully all eqs. from (2) to (21)!!!

-       P.9, Fig.6: Put reference;

-       P.10, Fig.7: How to ensure the galvanic separation of the sensors in fig. compared to the high potential of the line?

-       P.10, Fig.7: How is the operation of the sensors affected in the intense electromagnetic field? It is very possible that some of the sensors (analogue and digital) do not work. Have you checked their operation in real operating conditions (mounted on OPL)? Did you perform experiments only in laboratory conditions?

-       P.17, Section 4: You have compared the data with real experimental (from real OPL) measurements?

-       P.29, Section 5: An analysis was not made even under normal operating conditions of the OPL.

- In practice, safer, quicker, cheaper and more efficient methods can be implemented than the presented method.

Comments on the Quality of English Language

Minor editing of English language required.

Author Response

Dear reviewer, thanks for your recommendations.

            This article is part of a practical project in which the main beneficiary is the electricity distribution company. At the time of writing the article, laboratory tests have been carried out with good results and in the next period the system will be installed at the beneficiary's premises. The expected differences between laboratory tests on distances of less than 100 meters and in normal climatic conditions, compared to 100% real conditions, will accumulate over time and will highlight certain corrections due to extreme conditions of temperature, wind, rain and ice. Therefore in the final section it is proposed that: “In future work, as a result of collaboration with users, data will be collected both from the presented application and from direct measurements in extreme climatic conditions (wind and ice) in order to make the necessary corrections using software based on machine learning. This is because the mathematical model cannot accurately reproduce the sag determination of OPL cables in these contexts.”

            The complete software/hardware system also monitors and manages electrical parameters (such as electrical current and elements provided by a power quality analyzer). For reasons of reducing the complexity of the article, although a Hall split current transducer is shown in Figure 6, in the subsequent presentations (selections from the database, web server and IoT presentation), these parameters have not been highlighted because the article presentation focuses on the calculation of cable sags.

            We made the change in equation (1) and made all necessary revisions.

            Note that Figure 6 represents the architecture proposed by the authors in this project.

            We also point out that the software realizations are complete, and as can be seen, the contributions and novelties of the software are presented in the article.

            Please note that your questions are very clear, pertinent and come from a reviewer with practical achievements. That is why we hope you agree and understand that we cannot present the technical solutions in detail because they will be part of a patent proposal. We can only say that the power supply is done by a transformer based on a Rogowski cord that supplies the power to the sensors in the module placed on the power line. These are placed in a Faraday cage box. Some pictures that will not be included in the article, taken during the project, are shown below. This avoids interference and most of the modules used are low power. Most of the sensors are pole-mounted and powered by electricity from multiple sources including solar panels.

            In the paper presented, the line slope method has been used in principle, where based on the detection of the OPL cable slope angles, sag value can be obtained to which, in order to obtain an alternative verification relation, the measurement of the mechanical cable stress is also used, thus achieving increased accuracy and redundancy without significant burden on the hardware system. The table below summarises the methods, advantages and disadvantages of previous work.

Sag measurement method

Features

Detection method

This type of method is simple. It has a high degree of accuracy and the need for on-site operating staff excludes the possibility of real-time monitoring.

GPS

This type of method is accurate, it involves complex algorithms which are difficult to implement, it is costly.

Temperature and cable tension

This type of method is relatively inexpensive, it involves the measurement of too many parameters, the calculation algorithm is complex and it introduces large errors.

Projection technique

This type of method is easy to implement, it requires specialized photographers, the processing of images is difficult and it introduces large errors.

Line slope

This type of method is relatively simple, inexpensive, the calculation algorithm is simple and it introduces small errors.

            From an analytical point of view, starting from the differential equations, which are the basis of the sag measurement cathenary algorithm, in this paper two methods are combined in practice. The alternative methods Detection method, GPS and Projection technique, consist of direct measurements, which either cannot be considered in real time or introduce a high complexity to the algorithm, and a concrete comparison between the advantages and disadvantages of these methods are summarized in the table above. For a direct comparison between any of the methods you should actually implement all of them. This cannot be the subject of a single scientific article, so in this article the verification of the values obtained by the proposed algorithms is done in the laboratory experimental stage, by comparison with a direct measurement method (Detection method using a theodolite device for measurement).

Reviewer 3 Report

Comments and Suggestions for Authors

 The manuscript needs further development.

The following issues have to be argued in order to enhance the originality and readability of the MS.

1. What are the main questions that the work research focus. Here, the most of the MS is describing in an analytical way the experiment  and the process. Further analysis is suggested to the methods and the check of a equations described in order to clarify your experiments and results. For example eq (1) is not cleat about (starting from double expression about sinθ). Further validation is needed.

2. What is the literature gap to cover and the better results which they succeeded in? 

3. In the introduction a last paragraph is needed to describe the sections in a more comprehensive and precise way.

4. The results need a deeper approach and analysis of the figures in consistency and matching with the abstract, highlights and conclusion section, following the check and corrections as above (starting from sinθ). 

5.The extended discussion is missing and is suggested to compare the findings with the existing approaches. delignating the impacts and the novelty of the key findings of the research work, according to the journal's guidelines.

6. What is the difference to be added from this research work? Recent refs to compare the novelty would enhance the output and the readability

7. Nomenclature is required due to the large amount of abbreviations, which more times are not explained or cannot understood. An appendix about the methodology equations would be an appropriate solution.

Comments on the Quality of English Language

Language proofreading is needed.

Author Response

Dear reviewer, thanks for your recommendations.

  1. and 2. The introduction section presents the main contributions of this article. We believe that the first two points represent the essence of the novelty of this article:
  • Establishing alternative calculation methods and verification calculation relations for the OPL cable sag calculation depending on the groups of sensors used, thus obtaining a software method for detecting the failure of a sensor, a method whose usefulness can be expressed in terms of increasing system reliability;
  • Establishing the hardware and software architecture of the OPL online cable monitoring application.

            In the paper presented, the line slope method has been used in principle, where based on the detection of the OPL cable slope angles, sag value can be obtained to which, in order to obtain an alternative verification relation, the measurement of the mechanical cable stress is also used, thus achieving increased accuracy and redundancy without significant burden on the hardware system. The table below summarises the methods, advantages and disadvantages of previous work. 

Sag measurement method

Features

Detection method

This type of method is simple. It has a high degree of accuracy and the need for on-site operating staff excludes the possibility of real-time monitoring.

GPS

This type of method is accurate, it involves complex algorithms which are difficult to implement, it is costly.

Temperature and cable tension

This type of method is relatively inexpensive, it involves the measurement of too many parameters, the calculation algorithm is complex and it introduces large errors.

Projection technique

This type of method is easy to implement, it requires specialized photographers, the processing of images is difficult and it introduces large errors.

Line slope

This type of method is relatively simple, inexpensive, the calculation algorithm is simple and it introduces small errors.

  1. The rest of the article is structured as follows: Section 2 presents the cables and the calculation. The general architecture of the overhead power line cable monitoring system is presented in Section 3, and software applications for overhead power line cable monitoring system and cable sag calculation are presented in Section 4. Some conclusions and suggestions for further developments are presented in the final section.

  1. 5. and 6. From an analytical point of view, starting from the differential equations, which are the basis of the sag measurement cathenary algorithm, in this paper two methods are combined in practice. The alternative methods Detection method, GPS and Projection technique, consist of direct measurements, which either cannot be considered in real time or introduce a high complexity to the algorithm, and a concrete comparison between the advantages and disadvantages of these methods are summarized in the table above. For a direct comparison between any of the methods you should actually implement all of them. This cannot be the subject of a single scientific article, so in this article the verification of the values obtained by the proposed algorithms is done in the laboratory experimental stage, by comparison with a direct measurement method (Detection method using a theodolite device for measurement). In future work, as a result of collaboration with users, data will be collected both from the presented application and from direct measurements in extreme climatic conditions (wind and ice) in order to make the necessary corrections using software based on machine learning. This is because the mathematical model cannot accurately reproduce the sag determination of OPL cables in these contexts. This article is part of a practical project in which the main beneficiary is the electricity distribution company. At the time of writing the article, laboratory tests have been carried out with good results and in the next period the system will be installed at the beneficiary's premises. The expected differences between laboratory tests on distances of less than 100 meters and in normal climatic conditions, compared to 100% real conditions, will accumulate over time and will highlight certain corrections due to extreme conditions of temperature, wind, rain and ice.

            The complete software/hardware system also monitors and manages electrical parameters (such as electrical current and elements provided by a power quality analyzer). For reasons of reducing the complexity of the article, although a Hall split current transducer is shown in Figure 6, in the subsequent presentations (selections from the database, web server and IoT presentation), these parameters have not been highlighted because the article presentation focuses on the calculation of cable sags.

            We made the change in equation (1) and made all necessary revisions.

            Note that Figure 6 represents the architecture proposed by the authors in this project.

            We also point out that the software realizations are complete, and as can be seen, the contributions and novelties of the software are presented in the article.

            Please note that your questions are very clear, pertinent and come from a reviewer with practical achievements. That is why we hope you agree and understand that we cannot present the technical solutions in detail because they will be part of a patent proposal. We can only say that the power supply is done by a transformer based on a Rogowski cord that supplies the power to the sensors in the module placed on the power line. These are placed in a Faraday cage box. Some pictures that will not be included in the article, taken during the project, are shown below. This avoids interference and most of the modules used are low power. Most of the sensors are pole-mounted and powered by electricity from multiple sources including solar panels.

            We consider that the article is in accordance with the journal's guidelines. In support we insert the comments of reviewer 4: “The manuscript entitled “Real-time Monitoring of Cable Sag and Overhead Line Parameters Based on a Distributed Sensor Network and Implementation in a Web server and IoT” is well written and adequately structured. The manuscript describes the implementation of a hardware and software system designed for real-time monitoring of overhead power lines parameters and the calculation of the cable sag. The mathematical model based on the differential equations is explained as well as an algorithmic method for determining cable sag. The detail description of the hardware and software is given in the manuscript. My opinion is that manuscript is suitable for the publication in the journal with few minor corrections which can be done in editing phase”.

7. The nomenclature was inserted.

Nomenclature

OPL

Overhead Power Line

SCADA

Supervisory Control and Data Acquisition

LabVIEW

Laboratory Virtual Instrument Engineering Workbench

VI

Virtual Instrument

TDMS

Technical Data Management Streaming

WPAN

Wireless Personal Area Network

PAN

Personal Area Networks

RTU

Remote Terminal Unit

LabVIEW DSC

LabVIEW Datalogging and Supervisory Control

OPC

Open Platform Communications

API

Application Programming Interface

ODBC

Open Database Connectivity

MySQL

Relational database management system developed by Oracle that is based on structured query language

SMTP

Simple Mail Transfer Protocol

IP

Internet Protocol

DDNS

Dynamic Domain Name System

GSM

Global System for Mobile Communications

GPRS

General Packet Radio Service

RTCM

Radio Technical Commission for Maritime Services

NTRIP

Networked Transport of RTCM via Internet Protocol

HTTP

Hypertext Transfer Protocol

URL

Uniform Resource Locator

JSON

JavaScript Object Notation

HTML

HyperText Markup Language

REST

Representational State Transfer

IoT

Internet of Things

Reviewer 4 Report

Comments and Suggestions for Authors

The manuscript entitled “Real-time Monitoring of Cable Sag and Overhead Line Parameters Based on a Distributed Sensor Network and Implementation in a Web server and IoT” is well written and adequately structured. The manuscript describes the implementation of a hardware and software system designed for real-time monitoring of overhead power lines parameters and the calculation of the cable sag. The mathematical model based on the differential equations is explained as well as an algorithmic method for determining cable sag. The detail description of the hardware and software is given in the manuscript. My opinion is that manuscript is suitable for the publication in the journal with few minor corrections which can be done in editing phase:

·        Figure 2 shows formulas whose symbols have not been previously explained. It is necessary to provide an explanation of the labels or move Figure 2 after equation (4).

·        Figure 3 is unnecessary in my opinion, it burdens the manuscript.

Comments on the Quality of English Language

 English language fine.

Author Response

Dear reviewer, thanks for your recommendations.

            Thanks for the positive comments on the article as an entire.           

            We moved Figure 2 after equation (4) and hope that everything is OK.

            Since it is an open access journal and the number of pages is not limited, we consider that Figure 3 can be considered not so much as a burden but rather as a help to the reader interested in the numerical implementation in the form of an algorithm for the mathematical method to calculate the OPL cable sag. 

Round 2

Reviewer 2 Report

Comments and Suggestions for Authors

General observations:

-       In the title it should be mentioned: mechanical parameter;

-       In the title it should be mentioned: “in laboratory”, not for real OPL!

-       The article presents an interesting application, under laboratory conditions, but which faces major difficulties in practical implementation (on OPL, in real condition);

-       The work has a more theoretical character. How reliable would OPL be if so many sensors were installed between every two electricity pillars?

-       How ensure the galvanic separation of the sensors in fig. compared to the high potential of the line?

-       How is the operation of the sensors affected in the intense electromagnetic field? It is very possible that some of the sensors (analogue and digital) do not work. Have you checked their operation in real operating conditions (mounted on OPL)? Did you perform experiments only in laboratory conditions?

-       Section 4: You have compared the data with real experimental (from real OPL) measurements?

-       Section 5: An analysis was not made even under normal operating conditions of the OPL.

In practice, safer, quicker, cheaper and more efficient methods can be implemented than the presented method.

Comments on the Quality of English Language

Minor editing of English language required.

Author Response

Dear reviewer, thanks for your recommendations.

  • The complete software/hardware system also monitors and manages electrical parameters (such as electrical current and elements provided by a power quality analyzer). For reasons of reducing the complexity of the article, although a Hall split current transducer is shown in Figure 6, in the subsequent presentations (selections from the database, web server and IoT presentation), these parameters have not been highlighted because the article presentation focuses on the calculation of cable sags. Please note that the application is integrated into the Supervisory Control and Data Acquisition (SCADA) system of the beneficiary, and the parameters transmitted are not limited to the mechanical ones. Thus, we consider that it is not necessary to particularize the title of the article with "mechanical parameters".
  • The need to monitor and assess the condition of power transmission lines in real time as accurately as possible is obvious. The problem, then, is to develop specific and efficient calculation algorithms on the basis of the descriptive equations for calculating cable sag, and then, for real-time implementation, to use a set of suitable sensors together with a communication system that allows the data to be transmitted to a central server that performs the usual operations of displaying the state of the monitored line, but also integrates with the local Supervisory Control and Data Acquisition (SCADA) system. This article is part of a practical project in which the main beneficiary is the electricity distribution company. At the time of writing the article, laboratory tests have been carried out with good results and in the next period the system will be installed at the beneficiary's premises. The expected differences between laboratory tests on distances of less than 100 meters and in normal climatic conditions, compared to 100% real conditions, will accumulate over time and will highlight certain corrections due to extreme conditions of temperature, wind, rain and ice. Therefore in the final section it is proposed that: “In future work, as a result of collaboration with users, data will be collected both from the presented application and from direct measurements in extreme climatic conditions (wind and ice) in order to make the necessary corrections using software based on machine learning. This is because the mathematical model cannot accurately reproduce the sag determination of OPL cables in these contexts.” The software system is the same for use in the laboratory as for the beneficiary.

We are a national research and development institute with accredited laboratories with beneficiaries of our services on every continent except Australia. Accredited laboratories include low and high voltage, high power, mechanical forces and electromagnetic compatibility laboratories. At the following link:

https://www.inmrlaboratoryguide.com/listingcategory/high-wpower-laboratory/

you will find a site where the world's largest laboratories, high power and high voltage laboratories are grouped.

  • We hope you agree and understand that we cannot present the technical solutions in detail because they will be part of a patent proposal. We can only say that the power supply is done by a transformer based on a Rogowski cord that supplies the power to the sensors in the module placed on the power line. These are placed in a Faraday cage box. Some pictures that will not be included in the article, taken during the project, are shown below. This avoids interference and most of the modules used are low power. Most of the sensors are pole-mounted and powered by electricity from multiple sources including solar panels.

Still, as we said before, we cannot give all the technical details of the project. We mention that all the sensors and the two main boxes were also placed in the anechoic chamber for electromagnetic compatibility tests. We present as samples some pictures of the mechanical force (cable stress) sensor testing in our laboratory and some pictures of the Rogowski cord integration in the Faraday cage.

  • In the paper presented, the line slope method has been used in principle, where based on the detection of the OPL cable slope angles, sag value can be obtained to which, in order to obtain an alternative verification relation, the measurement of the mechanical cable stress is also used, thus achieving increased accuracy and redundancy without significant burden on the hardware system.

The table below summarises the methods, advantages and disadvantages of previous work.

Sag measurement method

Features

Detection method

This type of method is simple. It has a high degree of accuracy and the need for on-site operating staff excludes the possibility of real-time monitoring.

GPS

This type of method is accurate, it involves complex algorithms which are difficult to implement, it is costly.

Temperature and cable tension

This type of method is relatively inexpensive, it involves the measurement of too many parameters, the calculation algorithm is complex and it introduces large errors.

Projection technique

This type of method is easy to implement, it requires specialized photographers, the processing of images is difficult and it introduces large errors.

Line slope

This type of method is relatively simple, inexpensive, the calculation algorithm is simple and it introduces small errors.

From an analytical point of view, starting from the differential equations, which are the basis of the sag measurement cathenary algorithm, in this paper two methods are combined in practice. The alternative methods Detection method, GPS and Projection technique, consist of direct measurements, which either cannot be considered in real time or introduce a high complexity to the algorithm, and a concrete comparison between the advantages and disadvantages of these methods are summarized in the table above. For a direct comparison between any of the methods you should actually implement all of them. This cannot be the subject of a single scientific article, so in this article the verification of the values obtained by the proposed algorithms is done in the laboratory experimental stage, by comparison with a direct measurement method (Detection method using a theodolite device for measurement).

  • The system is one that transmits a series of electrical and mechanical parameters and integrates into the beneficiary's SCADA system. For example, the system also transmits power quality parameters and performs overhead power line ampacity estimation. The placement of the sensors shown in Figure 7 of the paper is done in two boxes, one on the electrical cable and one on the pole (only between two poles chosen by the beneficiary). The beneficiary chooses the two poles, usually as the pair with the most disadvantageous location in terms of ground level difference, with the highest possibility that in case of a rapid overload the line expansion will produce a short circuit (grounding). Therefore the sag value estimation of the OPL cables is a very important parameter that is monitored in real time.
